# PROTOTYPE-BASED OPTIMAL TRANSPORT FOR OUT-OF-DISTRIBUTION DETECTION

## ABSTRACT

Detecting Out-of-Distribution (OOD) inputs is crucial for improving the reliability of deep neural networks in the real-world deployment. In this paper, inspired by the inherent distribution shift between ID and OOD data, we propose a novel method that leverages optimal transport to measure the distribution discrepancy between test inputs and ID prototypes. The resulting transport costs are used to quantify the individual contribution of each test input to the overall discrepancy, serving as a desirable measure for OOD detection. To address the issue that solely relying on the transport costs to ID prototypes is inadequate for identifying OOD inputs closer to ID data, we generate virtual outliers to approximate the OOD region via linear extrapolation. By combining the transport costs to ID prototypes with the costs to virtual outliers, the detection of OOD data near ID data is emphasized, thereby enhancing the distinction between ID and OOD inputs. Experiments demonstrate the superiority of our method over state-of-the-art methods.

## 1 INTRODUCTION

Deep neural networks (DNNs) deployed in real-world scenarios often encounter out-of-distribution (OOD) inputs, such as inputs not belonging to one of the DNN's known classes. Ideally, reliable DNNs should be aware of what they do not know. However, they typically make overconfident predictions on OOD data (Nalisnick et al., 2018). This notorious behavior undermines the credibility of DNNs and could pose risks to involved users, particularly in safety-critical applications like autonomous driving (Filos et al., 2020) and biometric authentication (Wang & Deng, 2021). This gives rise to the importance of OOD detection, which identifies whether an input is OOD and enables conservative rejection or transferring decision-making to humans for better handling.

The representations of ID data within the same class tend to be gathered together after the training process, as shown in Figure 1(a). In contrast, the representations of OOD data are relatively far away from ID data, as they are not involved in the training process. In other words, the distributions of ID and OOD representations in the latent space exhibit a distinct separation. Therefore, we can expect that the distribution discrepancy between the representations of test inputs (i.e., a mixture of ID and OOD data) and pure ID data is primarily caused by the presence of OOD data. Such a distribution discrepancy motivates us to differentiate OOD data from test inputs by quantifying the individual contribution of each test input to the overall distribution discrepancy.

In this way, a critical question arises: *how to measure the distribution discrepancy between test inputs and ID data, while quantifying the contribution of each test input?* To this end, we utilize *optimal transport* (OT), a principled approach with rich geometric awareness for measuring the discrepancy between distributions. Concretely, OT aims to minimize the total transport cost between two distributions to measure the distribution discrepancy based on a predefined cost function (typically the geometric distance between samples). The smaller the total cost is, the closer the two distributions are. Since the total cost comprises the sum of transport costs between sample pairs, OT facilitates a fine-grained assessment of individual sample contributions to the overall discrepancy, making it particularly well-suited for OOD detection. Furthermore, the transport cost captures the geometric differences between ID and OOD representations in the latent space, providing a geometrically meaningful interpretation.

Based on the above intuition, in this paper, we propose a novel OOD detection method called **POT** that utilizes the **P**rototype-based **O**ptimal **T**ransport. Concretely, we first construct ID prototypes

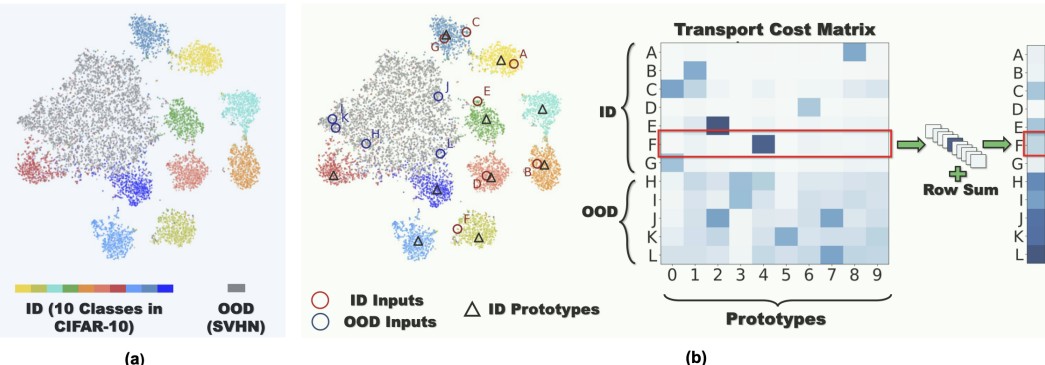

Figure 1: Illustration of our method for OOD detection. In (a), the representation distribution of OOD inputs is distinctly separated from ID inputs, visualized via t-SNE. The model is ResNet18 (He et al., 2016). The ID/OOD data is CIFAR-10 (Krizhevsky, 2009) and SVHN (Netzer et al., 2011). (b) shows a slice of the transport cost matrix, which is derived from the optimal transport between test inputs and ID prototypes (depicted as triangles). The row sum of a test input (labelled from A to L) represents the transport cost from it to all ID prototypes. Darker colors indicate higher transport costs. It is evident that the ID inputs (depicted as orange circles) generally incur lower transport costs compared to the OOD inputs (depicted as blue circles).

with the class-wise average of training sample representations to represent the distribution of ID data. We then apply OT between the representations of test inputs and the ID prototypes, obtaining a transport cost matrix where each entry indicates the transport cost between the corresponding pair of test input and prototype. As illustrated in Figure 1 (b), the total transport cost, calculated by summing all matrix entries, reflects the overall distribution discrepancy. The transport cost from each test sample to all ID prototypes (i.e., the row sum), serves as a measure of individual contribution to the overall discrepancy, indicating the likelihood of being OOD. However, the task of OOD detection remains inadequately addressed due to the presence of OOD data with smaller distribution shifts. These OOD data lie closer to ID data in the latent space, rendering the transport costs to ID prototypes insufficient for detecting them. To tackle this issue, we propose generating virtual outliers to approximate the OOD region, particularly the areas near ID prototypes, using linear extrapolation between ID prototypes and the average representation of test inputs. By integrating the transport costs from test inputs to ID prototypes with the cost to virtual outliers, the detection of OOD samples with smaller distribution shifts could be emphasized, thereby enhancing the overall distinction between ID and OOD data.

Our key contributions are as follows: (1) We present a novel perspective for OOD detection by measuring distribution discrepancy and propose an effective detection method using prototype-based optimal transport. (2) Extensive experiments on various benchmark datasets demonstrate that our proposed method achieves state-of-the-art (SOTA) performance, outperforming 21 previous OOD detection methods. Moreover, in the scenarios where training data is unavailable, our method consistently beats the robust competitors by a margin of 22.5% in FPR95 on the CIFAR-100 benchmark.

## 2 RELATED WORK

### 2.1 OOD DETECTION

OOD detection has attracted growing research attention in recent years. Existing approaches can generally be categorized into two major lines:

(1) One line of work utilizes the outputs from pretrained models to design scoring functions for differentiating OOD samples. These *post-hoc* methods can be further divided into three subcategories. 1) The confidence-based methods (Hendrycks & Gimpel, 2017; Sun et al., 2021; Song et al., 2022; Hendrycks et al., 2022; Wang et al., 2022b; Liu et al., 2023) adjusts model outputs to obtain the desired confidence, including maximum softmax probability (Hendrycks & Gimpel, 2017), energy (Liu et al., 2020), and generalized entropy (Liu et al., 2023). 2) The density-based methods (Hendrycks et al., 2022; Sun & Li, 2022; Zhang et al., 2023c; Liu et al., 2024) identifies certain properties or patterns of ID data, such as neuron coverage (Liu et al., 2024), by learning the corre-

sponding density functions, and the OOD samples that deviate from these properties or patterns tend to reside in low-density regions. 3) The distance-based methods (Lee et al., 2018; Ren et al., 2021; Sehwag et al., 2021; Sun et al., 2022) adopts distance metrics, e.g., Mahalanobis distance, between test input and ID samples or centroids differentiate OOD samples, under the assumption that OOD data lie relatively far away from ID data. Different from these works, we introduce a novel perspective for OOD detection on measuring distribution discrepancy. The most closely related work is dual divergence estimation (DDE) (Garg et al., 2023), which estimates the dual KL-Divergence between the test samples and ID samples. However, while DDE estimates divergence in a dual space optimized by DNNs and relies heavily on the quality of transformed sample representations, POT enables direct measurement of distribution discrepancy in the latent space.

(2) Another line of work focuses on altering models with training-time regularization to amplify the differences between OOD and ID samples (Hendrycks et al., 2019a; Chen et al., 2021; Ming et al., 2022; Zhang et al., 2023a; Wang et al., 2023; Lu et al., 2023). For example, by incorporating a suitable loss, models are encouraged to produce predictions with uniform distributions (Hendrycks et al., 2019a) or higher energy (Liu et al., 2020) for outlier data. Building on this, some approaches investigate refining or synthesizing outliers to improve the performance of models. For instance, Ming et al. (2022) utilizes a posterior sampling-based technique to select the most informative OOD samples from the large outlier set, while Wang et al. (2023) implicitly expands the auxiliary outliers by perturbing model parameters. However, the computational overhead of retraining can be prohibitive, especially when the parameter scale is large. Additionally, modifying models may also have side effects of degrading model performance on the original task. In contrast, this paper focus on post-hoc methods, which are easy to implement and generally applicable across different models. Such properties are highly practical for adopting OOD detection methods in real-world applications.

## 2.2 OPTIMAL TRANSPORT

As a mathematical tool for comparing distributions, optimal transport (OT) has been successfully employed in diverse machine learning tasks, including domain adaptation (Courty et al., 2016; Turrisi et al., 2022), generative adversarial training (Arjovsky et al., 2017), object detection (Ge et al., 2021), and partial-label learning (Wang et al., 2022a). The most related one to our work is (Lu et al., 2023), which also applies OT for the OOD detection problem. By assuming both an unlabeled and a labeled training set, Lu et al. (2023) uses OT to guide the clustering of samples for label assignment to the unlabeled samples, thereby augmenting the training data and facilitating the model retraining for OOD detection. In contrast, our work operates in a post-hoc manner without the requirement of extra training data or model retraining.

## 3 THE PROPOSED METHOD

### 3.1 PROBLEM SETTING

In the context of supervised multi-class classification, we denote the data space as $\mathcal{X}$ and the corresponding label space as $\mathcal{Y} = \{1, 2, \cdots, C\}$. The training dataset with in-distribution (ID) samples $\mathcal{D}_{tr} = \{(\boldsymbol{x}_i, y_i)\}_{i=1}^{n}$ is sampled from the joint distribution $P_{\mathcal{X}\mathcal{Y}}$. The marginal distribution on $\mathcal{X}$ is denoted as $P_{\mathcal{X}}^{in}$. The model trained on training data typically consists of a feature encoder $g : \mathcal{X} \to \mathcal{R}^d$, mapping the input $\boldsymbol{x} \in \mathcal{X}$ to a $d$-dimensional representation, and a linear classification layer $f : \mathcal{R}^d \to \mathcal{R}^C$, producing a logit vector containing classification confidence for each class. Given the test inputs $\mathcal{D}_{te} = \{\boldsymbol{x}_j\}_{j=1}^{m}$, the goal of OOD detection is to identify whether $\boldsymbol{x}_j$ is out-of-distribution w.r.t $P_{\mathcal{X}}^{in}$.

### 3.2 PROTOTYPE-BASED OPTIMAL TRANSPORT FOR OOD DETECTION

**Constructing Class Prototypes.** The key idea of our method is to use OT to measure the distribution discrepancy between test inputs and ID data while quantifying the individual contribution of each test input. A straightforward approach involves applying OT between test inputs and the training data. However, the standard OT is essentially a linear programming problem, suffering from cubic time complexity and incurring prohibitive computational cost when adopted to large-scale training sets (Peyré et al., 2019). An alternative is to sample a smaller subset of the training set for efficiency,

but this can result in missing classes, leading to mismatches in ID inputs. Such mismatches can exaggerate the contribution of ID inputs to the distribution discrepancy, thus incorrectly identifying them as OOD. To overcome this, we propose to characterize each class with a prototype and align test inputs to these prototypes. Specifically, given the training dataset $\mathcal{D}_{tr} = \{(\boldsymbol{x}_i, y_i)\}_{i=1}^n$, we construct each class prototype as the average representation for that class extracted from the feature encoder $g$:

$$\boldsymbol{\eta}_c = \frac{1}{N_c} \sum_{i=1}^n g(\boldsymbol{x}_i)\mathbf{1}\{y_i = c\}, \tag{1}$$

where $N_c$ is the number of samples in class $c$. The utilization of prototypes offers two key benefits: it reduces computational overhead by reducing the data scale and ensures all classes are represented, mitigating the risk of mismatches caused by missing classes.

**OOD Detection Using Prototype-based Optimal Transport.** By formalizing an optimal transport problem between the representations of test inputs $\{\boldsymbol{z}_j = g(\boldsymbol{x}_j)\}_{j=1}^m$ and ID prototypes $\{\boldsymbol{\eta}_i\}_{i=1}^C$, we search for the minimal transport cost that represents the distribution discrepancy, while subject to the mass conservation constraint:

$$\min_{\boldsymbol{\gamma} \in \Pi(\boldsymbol{\mu}, \boldsymbol{\nu})} \langle \boldsymbol{E}, \boldsymbol{\gamma} \rangle_F = \min_{\boldsymbol{\gamma} \in \Pi(\boldsymbol{\mu}, \boldsymbol{\nu})} \sum_{i=1}^C \sum_{j=1}^m E_{ij}\gamma_{ij} \tag{2}$$
$$\text{s.t. } \Pi(\boldsymbol{\mu}, \boldsymbol{\nu}) = \{\boldsymbol{\gamma} \in \mathcal{R}_+^{C \times m} | \boldsymbol{\gamma}\mathbf{1}_m = \boldsymbol{\mu}, \boldsymbol{\gamma}^T\mathbf{1}_C = \boldsymbol{\nu}\},$$

where $\langle \cdot, \cdot \rangle_F$ stands for the Frobenius dot-product. $\boldsymbol{\mu}$ and $\boldsymbol{\nu}$ are the probability simplexes of ID prototypes and test samples:

$$\boldsymbol{\mu} = \sum_{i=1}^C p_i \delta_{\boldsymbol{\eta}_i} \qquad \text{and} \qquad \boldsymbol{\nu} = \sum_{j=1}^m q_j \delta_{\boldsymbol{z}_j}, \tag{3}$$

where $p_i = \frac{N_i}{\sum_{i=1}^n N_i}$, $q_j = \frac{1}{m}$, and $\delta_{\boldsymbol{\eta}_i}$ is the Dirac at position $\boldsymbol{\eta}_i$. $\boldsymbol{\gamma}$ is a transport plan, essentially a joint probability matrix, with an entry $\gamma_{ij}$ describing the amount of probability mass transported from prototype $\boldsymbol{\eta}_i$ to test input $\boldsymbol{z}_j$. All feasible transport plans constitute the transportation polytope $\Pi(\boldsymbol{\mu}, \boldsymbol{\nu})$ (Cuturi, 2013). $\boldsymbol{E}$ is the ground cost matrix, where the entry $E_{ij}$ denotes the point-to-point moving cost between $\boldsymbol{\eta}_i$ and $\boldsymbol{z}_j$, which is defined with the Euclidean distance as $E_{ij} = ||\boldsymbol{\eta}_i - \boldsymbol{z}_j||_2$.

To efficiently resolve Equation 2, we introduce the entropic regularization term $H(\boldsymbol{\gamma})$ and express the optimization problem as:

$$\min_{\boldsymbol{\gamma} \in \Pi(\boldsymbol{\mu}, \boldsymbol{\nu})} \langle \boldsymbol{E}, \boldsymbol{\gamma} \rangle_F - \lambda H(\boldsymbol{\gamma}), \tag{4}$$

where $\lambda > 0$ and $H(\boldsymbol{\gamma}) = \sum_{i,j} \gamma_{ij}(\log \gamma_{ij} - 1)$ (Peyré et al., 2019). In this way, the optimal transport plan $\boldsymbol{\gamma}$ can be written as:

$$\boldsymbol{\gamma} = \text{Diag}(\boldsymbol{a})\boldsymbol{K}\text{Diag}(\boldsymbol{b}), \quad \text{where} \quad \boldsymbol{K} = \exp(-\boldsymbol{E}/\lambda). \tag{5}$$

Here, $\boldsymbol{a} \in \mathcal{R}^C$ and $\boldsymbol{b} \in \mathcal{R}^m$ are known as scaling variables. This formulation can be solved much faster using the Sinkhorn-Knopp algorithm (Cuturi, 2013):

$$\boldsymbol{a} \leftarrow \boldsymbol{\mu} \oslash (\boldsymbol{K}\boldsymbol{b}), \quad \boldsymbol{b} \leftarrow \boldsymbol{\nu} \oslash (\boldsymbol{K}^\top \boldsymbol{a}), \tag{6}$$

where $\oslash$ denotes element-wise division. Detailed derivations are provided in Appendix A. With the $\lambda$-strong convexity (Peyré et al., 2019), the entropic regularized OT could be solved in quadratic time complexity $O(nm)$, where $n$ and $m$ denote the number of data points in the two distributions, respectively. Since the ID distribution is represented with a fixed number of prototypes, the prototype-based OT has linear time complexity $O(Cm)$.

To evaluate whether an input sample $\boldsymbol{x}_j \in \mathcal{D}_{te}$ is OOD or not, we derive the cost of moving it to all ID prototypes $\mathcal{T}_j$ by decomposing the total transport cost:

$$\langle \boldsymbol{E}, \boldsymbol{\gamma} \rangle_F = \sum_{j=1}^m \sum_{i=1}^C E_{ij}\gamma_{ij} := \sum_{j=1}^m \mathcal{T}_j. \tag{7}$$

As the total transport cost serves as a measure of distributional discrepancy between ID and test data, a higher transport cost $\mathcal{T}$ for a test sample indicates a greater deviation from the ID data, suggesting that the sample is more likely to be OOD data. We emphasize that, while the solution of OT is the transport plan $\boldsymbol{\gamma}$, which is often the focus in many applications (Caron et al., 2020; Wang et al., 2022a), our concentration lies on the transport costs between sample pairs. Specifically, this is

Figure 2: Illustration of virtual outlier generation. The average representation of test inputs $\mathcal{M}$ lies between the average of ID inputs $\mathcal{M}_{\text{in}}$ and average of OOD inputs $\mathcal{M}_{\text{out}}$. We generate virtual outliers to approximate the OOD region using linear extrapolation between $\mathcal{M}$ and ID prototypes.

the product of the transport plan and the ground cost, which serves as a desirable and interpretable measure for OOD detection.

**Data Augmentation via Linear Extrapolation.** As described above, the transport cost can serve as a measure for OOD detection. However, relying solely on the cost to ID prototypes is insufficient for discerning OOD data with smaller distribution shifts from ID data. This is because such OOD data are located closer to ID data in the latent space and tend to incur lower transport costs to the ID prototypes. Meanwhile, the mass conservation constraint inherent in OT may exacerbate this issue by enforcing transportation between OOD data and prototypes when ID inputs are sparse. To address this issue, we propose to generate virtual outliers to approximate the OOD region using representation linear representation extrapolation. By integrating the transport costs from test inputs to both virtual outliers and ID prototypes, we introduce a contrastive transport cost, which enhances the detection performance, particularly for the OOD inputs with smaller distribution shifts.

Given two representations $z_i$ and $z_j$, the linear representation extrapolation is defined as:

$$z^* = z_i + \omega(z_j - z_i), \quad s.t. \quad \omega > 1 \vee \omega < 0. \tag{8}$$

Instead of generating outliers aimlessly by enumerating available sample pairs, we construct virtual outliers $\mathcal{P}^*$ by combining the prototypes $\mathcal{P}$ and the average representation of test inputs $\mathcal{M}$:

$$\mathcal{P}^* = \{\boldsymbol{\eta}_i^* = \boldsymbol{\eta}_i + \omega(\mathcal{M} - \boldsymbol{\eta}_i) + \boldsymbol{\eta}_i \in \mathcal{P}\}, \tag{9}$$

where $\omega > 1$ ensures that the generated points $\mathcal{P}^*$ lies beyond $\mathcal{M}$. The underlying intuition is that $\mathcal{M}$ can be expressed as a linear interpolation of the average representations of test ID samples $\mathcal{M}_{\text{in}}$ and test OOD samples $\mathcal{M}_{\text{out}}$:

$$\mathcal{M} := \frac{N_{\text{in}}}{N}\mathcal{M}_{\text{in}} + \frac{N - N_{\text{in}}}{N}\mathcal{M}_{\text{out}}, \tag{10}$$

where $N_{\text{in}}$ and $N_{\text{out}}$ denote the number of test ID samples and total test samples, respectively. As illustrated in Figure 2, the point $\mathcal{M}$ resides between ID and OOD data, guiding the direction for outlier generation in response to distribution shifts. As parameter $\omega$ increases, the generated virtual outliers progressively move away from the ID prototypes towards the OOD region in the latent space. By choosing an appropriate parameter, we can control the location of generated virtual outliers to emphasise the detection of OOD inputs with smaller distributions shifts. In contrast to the method (Zhu et al., 2023) that conducts informative extrapolation to synthesize numerous outliers during training with assumed auxiliary outliers, the linear representation extrapolation is a lightweight operation that does not require training or auxiliary outliers.

Likewise, after generating the virtual outliers $\mathcal{P}^*$, we apply the entropic regularized OT between $\mathcal{P}^*$ and the test inputs to obtain the corresponding transport cost $\mathcal{T}^*$. By taking the difference of transport costs from test inputs to the ID prototypes and to the virtual outliers, we derive the contrastive transport cost $\mathcal{T} - \mathcal{T}^*$ as the final OOD score. Given the opposite trends of $\mathcal{T}$ and $\mathcal{T}^*$ in indicating whether a sample is ID or OOD, a higher contrastive transport cost denotes a higher likelihood of the test input being OOD.

# 4 EXPERIMENTS

In this section, we preform extensive experiments over OOD detection benchmarks. All the experimental setup adheres to the latest version of OpenOOD, an open repository for benchmarking generalized OOD detection[1] (Yang et al., 2022; Zhang et al., 2023b).

---

[1]https://github.com/Jingkang50/OpenOOD.

## 4.1 COMMON SETUP

**Datasets.** We assess the performance of our proposed **POT** on the widely used CIFAR-100 benchmark and ImageNet-1k benchmark, which regard CIFAR-100 and ImageNet-1k as ID datasets, respectively. For CIFAR-100, we use the standard split, with 50,000 training images and 10,000 test images. For ImageNet-1k, we utilize 50,000 images from the validation set as ID test set. For each benchmark, OpenOOD splits OOD data to Far-OOD and Near-OOD based on their degrees of the semantic similarity with ID data. Specifically, the CIFAR-100 benchmark utilizes four Far-OOD datasets: MNIST (Deng, 2012), SVHN (Netzer et al., 2011), Textures (Cimpoi et al., 2014), and Places365 (Zhou et al., 2018), along with two Near-OOD datasets: CIFAR-10 and Tiny Imagenet (Le & Yang, 2015). For the large-scale ImageNet-1k benchmark, it considers iNaturalist (Horn et al., 2018), Textures (Cimpoi et al., 2014), and OpenImage-O (Wang et al., 2022b) as Far-OOD datasets. In terms of Near-OOD datasets, SSB-hard (Vaze et al., 2022) and NINCO (Bitterwolf et al., 2023) are included.

**Models.** For the CIFAR-100 benchmark, we utilize ResNet18 (He et al., 2016) as the model backbone, which is trained on the ID training samples for 100 epochs. We evaluate OOD detection methods over three checkpoints. For the ImageNet-1k benchmark, we employ ResNet50 and ViT-b16 (Dosovitskiy et al., 2021) models pretrained on ImageNet-1k and use the official checkpoints from PyTorch. For more training details, please refer to OpenOOD.

**Baselines.** Since POT performs in a *post-hoc* manner, we primarily compare against 21 *post-hoc* OOD detection methods, including OpenMax (Bendale & Boult, 2016), MSP (Hendrycks & Gimpel, 2017), ODIN (Liang et al., 2018), MDS (Lee et al., 2018), MDSEns (Lee et al., 2018), RMDS (Ren et al., 2021), Gram (Sastry & Oore, 2020), EBO (Liu et al., 2020), GradNorm (Huang et al., 2021), ReAct (Sun et al., 2021), MLS (Hendrycks et al., 2022), KLM (Hendrycks et al., 2022), VIM (Wang et al., 2022b), KNN (Sun et al., 2022), DICE (Sun & Li, 2022), RankFeat (Song et al., 2022), ASH (Djurisic et al., 2023), SHE (Zhang et al., 2023c), GEN (Liu et al., 2023), DDE (Garg et al., 2023), NAC-UE (Liu et al., 2024). The results for DDE are reproduced using the official codebase [2] while the results for the remaining methods are sourced from the implementations in OpenOOD.

**Hyperparameter tuning.** In line with OpenOOD, we use ID and OOD validation sets for hyperparameter selection. Specifically, for the CIFAR-100 benchmark, 1,000 images are held out from the ID test set as the ID validation set, and 1,000 images spanning 20 categories from Tiny ImageNet (Le & Yang, 2015) are reserved as the OOD validation set. For the ImageNet-1k benchmark, 5,000 images from the ID test set and 1,763 images from OpenImage-O are held out for the ID and OOD validation sets, respectively. Please note that all the validation samples are disjoint with the test samples. The validation sets are used to tune hyperparameters, including the entropic regularization coefficient $\lambda$ and the linear extrapolation parameter $\omega$ for POT. Please refer to the Appendix B for more details.

**Evaluation metrics.** We report the following widely adopted metrics: (1) area under the receiver operating characteristic curve (AUROC); (2) false positive rate of OOD samples when the true positive rate of ID samples is equal to $95\%$ (FPR95).

**Implementation details.** During the test phase, the test set comprises samples from both the ID test set and OOD dataset. We consider the assumption of having access to the entire test set to be overly restrictive. To this end, we relax the assumption by allowing test inputs arriving in batches, where random batch division is applied to the test set. The default test batch size is set to 512 and we also include ablation studies with varying batch sizes. For each test batch, we employ POT to calculate transport cost scores for the samples. After aggregating the scores across all test samples, we calculate the evaluation metrics for comparison.

## 4.2 EMPIRICAL RESULTS AND ANALYSIS

**Main results.** The results for Far-OOD and Near-OOD detection on the CIFAR-100 benchmark are presented in Table 1 and Table 2, respectively. Our proposed POT consistently obtains either the best or second-best results across all datasets and OOD detection metrics. Specifically, on the Far-OOD track, POT achieves significant reductions in average FPR95, with decreases of 25.6% and 8.87%

---

[2]https://github.com/morganstanley/MSML/tree/main/papers/OOD_Detection_via_Dual_Divergence_Estimation

Table 1: Far-OOD detection performance on CIFAR-100 benchmark. ↑ denotes the higher value is better, while ↓ indicates lower values are better. We format **first** and second results.

| Method | MINIST | | SVHN | | Textures | | Places365 | | Average | |
|---|---|---|---|---|---|---|---|---|---|---|
| | FPR95↓ | AUROC↑ | FPR95↓ | AUROC↑ | FPR95↓ | AUROC↑ | FPR95↓ | AUROC↑ | FPR95↓ | AUROC↑ |
| OpenMax | $53.97_{\pm4.71}$ | $75.89_{\pm1.40}$ | $52.81_{\pm1.89}$ | $82.05_{\pm1.55}$ | $56.16_{\pm1.86}$ | $80.46_{\pm0.10}$ | $54.99_{\pm1.42}$ | $79.22_{\pm0.41}$ | $54.48_{\pm0.63}$ | $79.40_{\pm0.41}$ |
| MSP | $57.24_{\pm4.67}$ | $76.08_{\pm1.86}$ | $58.43_{\pm2.61}$ | $78.68_{\pm0.95}$ | $61.79_{\pm1.31}$ | $77.32_{\pm0.71}$ | $56.64_{\pm0.87}$ | $79.22_{\pm0.29}$ | $58.52_{\pm1.12}$ | $77.83_{\pm0.45}$ |
| ODIN | $45.93_{\pm3.24}$ | $83.79_{\pm1.30}$ | $67.21_{\pm3.95}$ | $74.72_{\pm0.77}$ | $62.39_{\pm2.87}$ | $79.34_{\pm1.08}$ | $59.73_{\pm0.86}$ | $79.45_{\pm0.25}$ | $58.81_{\pm0.78}$ | $79.32_{\pm0.22}$ |
| MDS | $71.70_{\pm2.89}$ | $67.47_{\pm0.81}$ | $67.72_{\pm6.05}$ | $70.20_{\pm6.52}$ | $70.55_{\pm2.50}$ | $76.23_{\pm0.69}$ | $79.57_{\pm0.34}$ | $63.17_{\pm0.50}$ | $72.38_{\pm1.53}$ | $69.27_{\pm1.41}$ |
| MDSEns | $2.86_{\pm0.85}$ | $98.20_{\pm0.78}$ | $82.57_{\pm2.57}$ | $53.74_{\pm1.62}$ | $84.91_{\pm0.87}$ | $69.75_{\pm1.14}$ | $96.58_{\pm0.19}$ | $42.32_{\pm0.74}$ | $66.73_{\pm1.05}$ | $66.00_{\pm0.69}$ |
| RMDS | $51.99_{\pm6.34}$ | $79.78_{\pm2.50}$ | $51.10_{\pm3.62}$ | $85.09_{\pm1.09}$ | $54.06_{\pm1.02}$ | $83.61_{\pm0.52}$ | $53.58_{\pm0.33}$ | $83.39_{\pm0.47}$ | $52.68_{\pm0.65}$ | $82.97_{\pm0.42}$ |
| Gram | $53.35_{\pm7.51}$ | $80.78_{\pm4.14}$ | $20.40_{\pm1.69}$ | $95.47_{\pm0.58}$ | $89.84_{\pm2.87}$ | $70.61_{\pm1.44}$ | $95.03_{\pm0.63}$ | $46.09_{\pm1.28}$ | $64.66_{\pm2.30}$ | $73.24_{\pm1.05}$ |
| EBO | $52.62_{\pm3.83}$ | $79.18_{\pm1.36}$ | $53.19_{\pm3.25}$ | $82.28_{\pm1.78}$ | $62.38_{\pm2.08}$ | $78.35_{\pm0.84}$ | $57.70_{\pm0.87}$ | $79.50_{\pm0.23}$ | $56.47_{\pm1.41}$ | $79.83_{\pm0.62}$ |
| GradNorm | $86.96_{\pm1.45}$ | $65.35_{\pm1.12}$ | $69.38_{\pm8.40}$ | $77.23_{\pm4.88}$ | $92.37_{\pm0.58}$ | $64.58_{\pm0.13}$ | $85.41_{\pm0.39}$ | $69.66_{\pm0.17}$ | $83.53_{\pm2.01}$ | $69.20_{\pm1.08}$ |
| ReAct | $56.03_{\pm5.67}$ | $78.37_{\pm1.59}$ | $49.89_{\pm1.95}$ | $83.25_{\pm1.00}$ | $55.02_{\pm0.81}$ | $80.15_{\pm0.46}$ | $55.34_{\pm0.49}$ | $80.01_{\pm0.11}$ | $54.07_{\pm1.57}$ | $80.45_{\pm0.50}$ |
| MLS | $52.94_{\pm3.83}$ | $78.91_{\pm1.47}$ | $53.43_{\pm3.22}$ | $81.90_{\pm1.53}$ | $62.37_{\pm2.16}$ | $78.39_{\pm0.84}$ | $57.64_{\pm0.92}$ | $79.74_{\pm0.24}$ | $56.60_{\pm1.41}$ | $79.73_{\pm0.58}$ |
| KLM | $72.88_{\pm6.56}$ | $74.15_{\pm2.60}$ | $50.32_{\pm7.06}$ | $79.49_{\pm0.47}$ | $81.88_{\pm5.87}$ | $75.75_{\pm0.48}$ | $81.60_{\pm1.37}$ | $75.68_{\pm0.26}$ | $71.67_{\pm2.07}$ | $76.27_{\pm0.53}$ |
| VIM | $48.34_{\pm1.03}$ | $81.84_{\pm1.03}$ | $46.28_{\pm5.52}$ | $82.89_{\pm3.78}$ | $46.84_{\pm2.28}$ | $85.90_{\pm0.79}$ | $61.64_{\pm0.70}$ | $75.85_{\pm0.36}$ | $50.77_{\pm0.98}$ | $81.62_{\pm0.62}$ |
| KNN | $48.59_{\pm4.66}$ | $82.36_{\pm1.54}$ | $51.43_{\pm3.15}$ | $84.26_{\pm1.11}$ | $53.56_{\pm2.35}$ | $83.66_{\pm0.84}$ | $60.80_{\pm0.92}$ | $79.42_{\pm0.47}$ | $53.59_{\pm0.25}$ | $82.43_{\pm0.17}$ |
| DICE | $51.80_{\pm3.68}$ | $79.86_{\pm1.89}$ | $48.96_{\pm3.34}$ | $84.45_{\pm2.04}$ | $64.23_{\pm1.59}$ | $77.63_{\pm0.34}$ | $59.43_{\pm1.20}$ | $78.31_{\pm0.66}$ | $56.10_{\pm0.62}$ | $80.06_{\pm0.19}$ |
| RankFeat | $75.02_{\pm5.82}$ | $63.03_{\pm3.85}$ | $58.17_{\pm2.07}$ | $72.37_{\pm1.51}$ | $66.90_{\pm3.79}$ | $69.40_{\pm3.09}$ | $77.42_{\pm1.93}$ | $63.81_{\pm1.83}$ | $69.38_{\pm1.10}$ | $67.15_{\pm1.49}$ |
| ASH | $66.60_{\pm3.88}$ | $77.23_{\pm0.46}$ | $45.51_{\pm2.82}$ | $85.76_{\pm1.38}$ | $61.34_{\pm2.83}$ | $80.72_{\pm0.71}$ | $62.89_{\pm1.08}$ | $78.75_{\pm0.16}$ | $59.09_{\pm2.53}$ | $80.61_{\pm0.66}$ |
| SHE | $58.82_{\pm2.75}$ | $76.72_{\pm1.08}$ | $58.60_{\pm7.63}$ | $81.22_{\pm4.05}$ | $73.34_{\pm3.35}$ | $73.65_{\pm1.29}$ | $65.23_{\pm0.86}$ | $76.29_{\pm0.52}$ | $64.00_{\pm2.73}$ | $76.97_{\pm1.17}$ |
| GEN | $54.81_{\pm4.80}$ | $78.09_{\pm1.82}$ | $56.14_{\pm2.17}$ | $81.24_{\pm1.05}$ | $61.13_{\pm1.49}$ | $78.70_{\pm0.80}$ | $56.07_{\pm0.78}$ | $80.31_{\pm0.22}$ | $57.04_{\pm1.01}$ | $79.59_{\pm0.54}$ |
| DDE | $\mathbf{0.01}_{\pm0.01}$ | $\mathbf{99.93}_{\pm0.02}$ | $\mathbf{0.23}_{\pm0.03}$ | $\mathbf{99.31}_{\pm0.09}$ | $40.30_{\pm1.24}$ | $\underline{93.13}_{\pm0.29}$ | $\underline{52.34}_{\pm0.61}$ | $\underline{88.21}_{\pm0.23}$ | $\underline{23.22}_{\pm0.45}$ | $\mathbf{95.14}_{\pm0.15}$ |
| NAC-UE | $21.44_{\pm5.22}$ | $93.24_{\pm1.33}$ | $24.23_{\pm3.88}$ | $92.43_{\pm1.03}$ | $\underline{40.19}_{\pm1.97}$ | $89.34_{\pm0.56}$ | $73.93_{\pm1.52}$ | $72.92_{\pm0.78}$ | $39.95_{\pm1.36}$ | $86.98_{\pm0.26}$ |
| **POT** | $\underline{0.98}_{\pm0.08}$ | $\underline{99.73}_{\pm0.02}$ | $\underline{2.13}_{\pm0.21}$ | $\underline{99.39}_{\pm0.03}$ | $\mathbf{25.56}_{\pm3.93}$ | $\mathbf{95.28}_{\pm0.44}$ | $\mathbf{28.74}_{\pm0.22}$ | $\mathbf{92.42}_{\pm0.12}$ | $\mathbf{14.35}_{\pm1.06}$ | $\underline{96.70}_{\pm0.08}$ |

Table 2: Near-OOD detection performance on CIFAR-100 benchmark. ↑ denotes the higher value is better, while ↓ indicates lower values are better.

| Method | CIFAR-10 | | Tiny ImageNet | | Average | |
|---|---|---|---|---|---|---|
| | FPR95↓ | AUROC↑ | FPR95↓ | AUROC↑ | FPR95↓ | AUROC↑ |
| OpenMax | $60.19_{\pm0.87}$ | $74.34_{\pm0.33}$ | $52.79_{\pm0.43}$ | $78.48_{\pm0.09}$ | $56.49_{\pm0.58}$ | $76.41_{\pm0.20}$ |
| MSP | $58.90_{\pm0.93}$ | $78.47_{\pm0.07}$ | $50.78_{\pm0.57}$ | $81.96_{\pm0.20}$ | $54.84_{\pm0.58}$ | $80.21_{\pm0.13}$ |
| ODIN | $60.61_{\pm0.52}$ | $78.18_{\pm0.14}$ | $55.28_{\pm0.45}$ | $81.53_{\pm0.10}$ | $57.95_{\pm0.45}$ | $79.86_{\pm0.11}$ |
| MDS | $88.01_{\pm0.51}$ | $55.89_{\pm0.22}$ | $78.68_{\pm1.48}$ | $61.83_{\pm0.19}$ | $83.35_{\pm0.76}$ | $58.86_{\pm0.09}$ |
| MDSEns | $95.94_{\pm0.15}$ | $43.85_{\pm0.31}$ | $95.76_{\pm0.13}$ | $49.14_{\pm0.22}$ | $95.85_{\pm0.05}$ | $46.49_{\pm0.25}$ |
| RMDS | $61.36_{\pm0.23}$ | $77.77_{\pm0.21}$ | $49.50_{\pm0.58}$ | $82.58_{\pm0.02}$ | $55.43_{\pm0.29}$ | $80.18_{\pm0.10}$ |
| Gram | $92.69_{\pm0.58}$ | $49.41_{\pm0.54}$ | $92.34_{\pm0.84}$ | $53.12_{\pm1.66}$ | $92.51_{\pm0.39}$ | $51.26_{\pm0.80}$ |
| EBO | $59.19_{\pm0.74}$ | $79.05_{\pm0.10}$ | $52.36_{\pm0.59}$ | $82.58_{\pm0.08}$ | $55.77_{\pm0.64}$ | $80.82_{\pm0.09}$ |
| GradNorm | $84.30_{\pm0.38}$ | $70.32_{\pm0.20}$ | $87.30_{\pm0.59}$ | $69.58_{\pm0.79}$ | $85.80_{\pm0.46}$ | $69.95_{\pm0.47}$ |
| ReAct | $61.29_{\pm0.43}$ | $78.65_{\pm0.05}$ | $51.64_{\pm0.41}$ | $82.72_{\pm0.08}$ | $56.47_{\pm0.42}$ | $80.69_{\pm0.06}$ |
| MLS | $59.10_{\pm0.63}$ | $79.21_{\pm0.10}$ | $52.19_{\pm0.42}$ | $82.74_{\pm0.08}$ | $55.64_{\pm0.52}$ | $80.97_{\pm0.09}$ |
| KLM | $84.77_{\pm2.99}$ | $73.92_{\pm0.23}$ | $71.59_{\pm0.79}$ | $79.16_{\pm0.30}$ | $78.18_{\pm1.30}$ | $76.54_{\pm0.24}$ |
| VIM | $70.63_{\pm0.44}$ | $72.21_{\pm0.42}$ | $54.54_{\pm0.31}$ | $77.87_{\pm0.13}$ | $62.59_{\pm0.26}$ | $75.04_{\pm0.14}$ |
| KNN | $72.82_{\pm0.50}$ | $77.01_{\pm0.26}$ | $49.63_{\pm0.61}$ | $83.31_{\pm0.16}$ | $61.22_{\pm0.15}$ | $80.16_{\pm0.15}$ |
| DICE | $60.98_{\pm1.10}$ | $78.04_{\pm0.32}$ | $55.36_{\pm0.59}$ | $80.50_{\pm0.25}$ | $58.17_{\pm0.50}$ | $79.27_{\pm0.22}$ |
| RankFeat | $82.78_{\pm1.56}$ | $58.04_{\pm2.36}$ | $78.37_{\pm1.09}$ | $65.63_{\pm0.24}$ | $80.57_{\pm1.11}$ | $61.84_{\pm1.29}$ |
| ASH | $68.06_{\pm0.41}$ | $76.47_{\pm0.30}$ | $63.47_{\pm1.10}$ | $79.79_{\pm0.24}$ | $65.77_{\pm0.49}$ | $78.13_{\pm0.17}$ |
| SHE | $60.47_{\pm0.58}$ | $78.13_{\pm0.02}$ | $58.42_{\pm0.76}$ | $79.52_{\pm0.33}$ | $59.45_{\pm0.34}$ | $78.83_{\pm0.17}$ |
| GEN | $58.65_{\pm0.92}$ | $79.40_{\pm0.06}$ | $49.82_{\pm0.29}$ | $83.15_{\pm0.15}$ | $54.23_{\pm0.54}$ | $81.27_{\pm0.10}$ |
| DDE | $62.35_{\pm2.12}$ | $81.32_{\pm0.28}$ | $61.20_{\pm2.11}$ | $80.34_{\pm0.95}$ | $61.78_{\pm2.11}$ | $80.83_{\pm0.60}$ |
| NAC-UE | $80.84_{\pm1.38}$ | $71.92_{\pm0.77}$ | $62.78_{\pm1.69}$ | $79.43_{\pm0.45}$ | $71.81_{\pm1.51}$ | $75.67_{\pm0.56}$ |
| **POT** | $\mathbf{41.63}_{\pm2.28}$ | $\mathbf{87.51}_{\pm0.44}$ | $\mathbf{46.94}_{\pm0.43}$ | $\mathbf{85.50}_{\pm0.04}$ | $\mathbf{44.28}_{\pm1.26}$ | $\mathbf{86.51}_{\pm0.20}$ |

compared to the previous leading baselines NAC-UE and DDE, respectively. While Near-OOD samples are considered more intractable to detect due to their similarity in semantic and style with ID samples, POT demonstrates an even greater performance advantage in detecting them, as shown in Table 2. For instance, POT surpasses the next best method, GEN, by 9.96% in average FPR95 and 5.22% in average AUROC. The results of the large-scale ImageNet-1k benchmark are shown in Table 3, where only the AUROC values are reported due to space limitation. As can be seen, POT also consistently outperforms all baseline methods in terms of average performance across different backbones and OOD datasets.

Table 3: OOD detection performance (AUROC↑) on ImageNet-1k. See Table 7 and Table 8 for full results.

| Backbone | Datasets | RMDS | ReAct | VIM | KNN | ASH | SHE | GEN | DDE | NAC-UE | **POT** |
|---|---|---|---|---|---|---|---|---|---|---|---|
| ViT-b16 | iNaturalist | 96.09 | 86.09 | 95.71 | 91.45 | 50.47 | 93.56 | 93.53 | 97.10 | 93.69 | **99.38** |
| | Openimage-O | 92.29 | 84.26 | 92.15 | 89.83 | 55.45 | 91.03 | 90.25 | 88.97 | 91.54 | **95.60** |
| | Textures | 89.38 | 86.69 | 90.61 | 91.12 | 47.87 | 92.67 | 90.25 | 88.96 | 94.17 | **95.36** |
| | Average (Far-OOD) | 92.59 | 85.68 | 92.82 | 90.80 | 51.26 | 92.42 | 91.34 | 91.68 | 93.13 | **96.78** |
| | SSB-hard | 72.79 | 63.24 | 69.34 | 65.93 | 54.12 | 68.11 | 70.19 | 76.29 | 68.04 | **80.39** |
| | NINCO | 87.28 | 75.45 | 84.63 | 82.22 | 53.07 | 84.16 | 82.47 | 81.32 | 82.45 | **87.51** |
| | Average (Near-OOD) | 80.03 | 69.34 | 76.98 | 74.08 | 53.59 | 76.13 | 76.33 | 78.81 | 75.25 | **83.95** |
| ResNet-50 | iNaturalist | 87.27 | 96.32 | 89.54 | 86.31 | 97.04 | 92.58 | 92.44 | 83.64 | 96.43 | **99.45** |
| | Openimage-O | 85.73 | 91.88 | 90.40 | 86.78 | 93.31 | 86.70 | 89.35 | 71.38 | 91.61 | **93.45** |
| | Textures | 86.07 | 92.80 | **97.96** | 97.06 | 96.91 | 93.63 | 87.63 | 86.84 | 97.77 | 95.58 |
| | Average (Far-OOD) | 86.36 | 93.67 | 92.63 | 90.05 | 95.76 | 90.97 | 89.81 | 80.62 | 95.27 | **96.16** |
| | SSB-hard | 71.47 | 73.09 | 65.10 | 61.78 | 73.11 | 71.83 | 72.11 | 56.52 | 68.21 | **83.37** |
| | NINCO | 82.22 | 81.71 | 78.54 | 79.41 | **83.37** | 76.42 | 81.71 | 62.93 | 81.16 | 78.13 |
| | Average (Near-OOD) | 76.84 | 77.40 | 71.82 | 70.59 | 78.24 | 74.12 | 76.91 | 59.73 | 74.68 | **80.75** |

Table 4: Results on the ImageNet-1k with different training methods. We employ the ResNet-50 as the backbone. Δ represents the subtraction results between the default CE scheme and other training schemes.

| Training | Baseline | | ASH | | NAC-UE | | POT | |
|---|---|---|---|---|---|---|---|---|
| | FPR95↓ | AUROC↑ | FPR95↓ | AUROC↑ | FPR95↓ | AUROC↑ | FPR95↓ | AUROC↑ |
| CE | - | - | 19.67 | 95.76 | 22.67 | 95.27 | 19.89 | 96.16 |
| ConfBranch | 51.17 | 83.97 | 78.77 | 72.64 | 32.21 | 93.63 | **20.34** | **95.90** |
| Δ | - | - | (+59.1) | (-23.12) | (+9.54) | (-1.64) | (+0.45) | (-0.26) |
| RotPred | 36.39 | 90.03 | 68.61 | 78.99 | 38.50 | 92.23 | **19.55** | **95.93** |
| Δ | - | - | (+48.94) | (-16.77) | (+15.83) | (-3.04) | (-0.34) | (-0.23) |
| GODIN | 51.03 | 85.50 | 57.06 | 88.07 | 29.44 | 93.99 | **22.74** | **95.25** |
| Δ | - | - | (+37.39) | (-7.69) | (+6.77) | (-1.28) | (+2.85) | (-0.91) |

**Integration with training methods.** In the other line of work for OOD detection, training methods employ retraining strategies with training-time regularization to provide a modified model. A important property of *post-hoc* methods is that they are applicable to different model architectures and training losses. To this end, we examine the performance of *post-hoc* methods when integrated with established training methods. We evaluate on the Far-OOD track of the ImageNet-1k benchmark using ResNet-50 as the backbone, comparing POT with ASH and NAC-UE, which have achieved top results on this benchmark (see Table 3). Consistent with the experimental setup of NAC-UE, we utilize three training schemes: ConfBranch (DeVries & Taylor, 2018), RotPred (Hendrycks et al., 2019b), and GODIN (Hsu et al., 2020), with softmax cross-entropy (CE) loss as the default training scheme for comparison. We report the average results in Table 4, where *Baseline* refers to the detection method employed in the original paper and Δ denotes the subtraction results between the default CE scheme and other training schemes. According to the results, POT has remarkable improvements on the baseline methods, while outperforming ASH and NAC-UE. Importantly, POT maintains stable performance across three training schemes, whereas ASH and NAC-UE both exhibit notable performance degradation compared to the default CE training scheme. Such results demonstrates that POT is generic to be seamlessly integrated with different training methods.

**What if training data is unavailable.** Although existing *post-hoc* methods can be applied to pretrained models without cumbersome retraining, many require access to at least a portion of training data (Liu et al., 2023). Considering some scenarios where training data is unavailable, such as commercial data involving privacy, some approaches further explore OOD detection without the requirement for training data such as ASH and GEN. To make POT applicable to such scenarios, we draw inspiration from the work (Tanwisuth et al., 2021) and construct the class prototypes with the neural network weights $\mathbf{W} \in \mathcal{R}^{d \times C}$ of the classification layer $f$. Each column of the weight matrix $\mathbf{W}$ corresponds to a $d$-dimensional class prototype. The underlying idea is that the process of learning class prototypes with learnable parameters in the latent space, is closely similar to the training process of the linear classification layer. In other words, obtaining prototypes in this way does not require any training data and additional parameters.

Table 5: OOD detection performance (AUROC↑) of methods without the requirement for training data.

| Method | CIFAR-100 | | | | ImageNet-1k | | | |
| | Far-OOD | | Near-OOD | | Far-OOD | | Near-OOD | |
| | FPR95↓ | AUROC↑ | FPR95↓ | AUROC↑ | FPR95↓ | AUROC↑ | FPR95↓ | AUROC↑ |
|---|---|---|---|---|---|---|---|---|
| MSP | 58.53 | 77.83 | 54.84 | 80.21 | 51.77 | 86.03 | 81.56 | 73.56 |
| ODIN | 58.81 | 79.32 | 57.95 | 79.86 | 86.04 | 76.08 | 90.76 | 64.45 |
| EBO | 56.47 | 79.83 | 55.77 | 80.82 | 85.34 | 78.99 | 93.09 | 62.60 |
| GradNorm | 83.53 | 69.20 | 85.80 | 69.95 | 92.60 | 41.75 | 94.68 | 39.45 |
| ReAct | 54.07 | 80.45 | 56.47 | 80.69 | 53.96 | 85.68 | 84.42 | 69.34 |
| MLS | 56.60 | 79.73 | 55.64 | 80.97 | 78.91 | 83.55 | 92.06 | 68.42 |
| RankFeat | 69.38 | 67.15 | 80.57 | 61.84 | / | / | / | / |
| ASH | 59.09 | 80.61 | 65.77 | 78.13 | 96.69 | 51.26 | 95.07 | 53.60 |
| GEN | 57.04 | 79.59 | 54.23 | 81.27 | 32.16 | 91.34 | 70.66 | 76.33 |
| **POT** | **21.26** | **94.27** | **45.02** | **84.76** | **29.40** | **93.35** | **65.82** | **80.42** |

To verify the effectiveness of POT in such scenarios, we compare it to the baseline methods not requiring access to the training dataset. We report the average AUROC for both Far-OOD and Near-OOD datasets in Table 5. The results show that POT continues to outperform all competitors across all metrics. Notably, compared to GEN, POT significantly reduces the average FPR95 by 35.78% on the Far-OOD datasets and by 9.21% on the Near-OOD datasets of the CIFAR-100 benchmark.

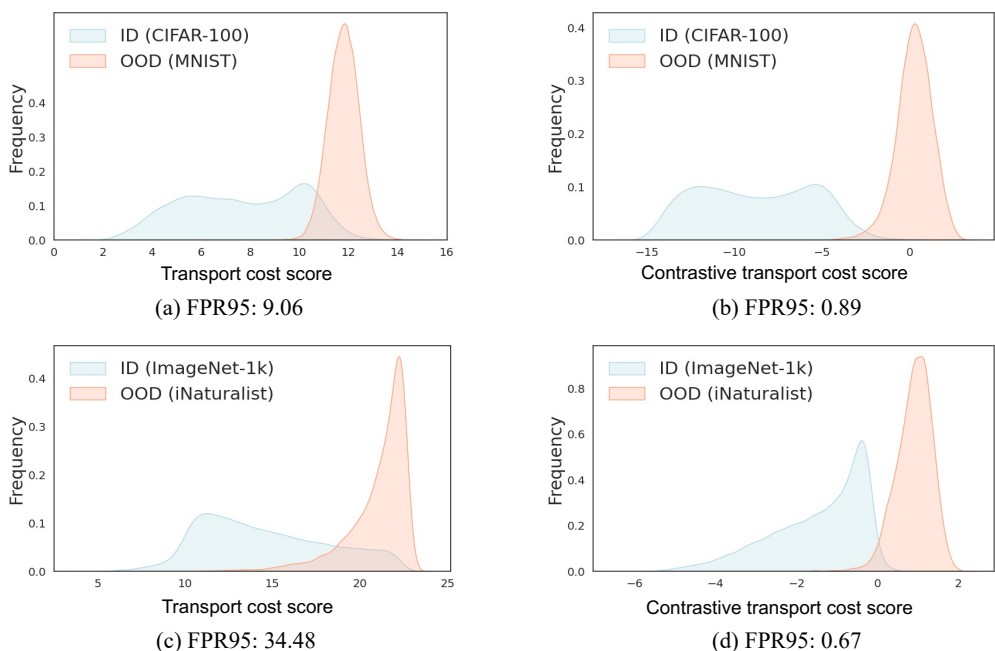

(a) FPR95: 9.06

(b) FPR95: 0.89

(c) FPR95: 34.48

(d) FPR95: 0.67

Figure 3: Ablation study on the effect of virtual outliers. We contrast the distribution for the transport cost score without virtual outliers (a & c) and the contrastive transport cost score with virtual outliers (b & d). The used models are ResNet-18 for CIFAR-100 and ViT-b16 for ImageNet-1k, respectively. The introduction of virtual outliers makes a more distinguishable score, leading to enhanced OOD detection performance.

### 4.3 ABLATION STUDY AND ANALYSIS.

**Ablation on virtual outliers.** In this ablation, we compare the OOD detection performance of POT with and without virtual outliers, which are generated to approximate the OOD region using linear representation extrapolation. Figure 3 displays the results, where we visualize the data distribution in the CIFAR-100 and ImageNet-1k benchmarks. Using the virtual outliers leads to clearer separability between ID and OOD samples, whereas the transport cost score without virtual out-

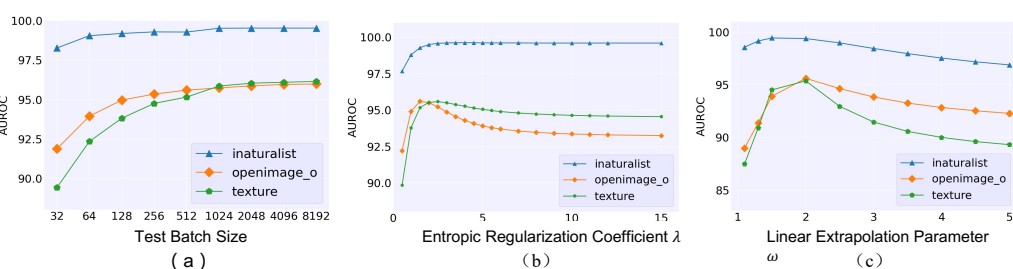

Figure 4: Ablation across different parameters in POT including: (a) test batch size; (b) entropic regularization coefficient $\lambda$ of OT; (c) linear extrapolation parameter $\omega$ in generating virtual outlier.

liers exhibits larger overlap. The results show that the introduction of virtual outliers enhances the distinguishability between ID and OOD, resulting in more effective OOD detection.

**Parameter analysis.** We ablate along individual parameters with ImageNet-1k as ID data in Figure 4, where the ViT-b16 is utilized as the backbone for analysis. In Figure 4 (a), we find that increasing the test batch size is beneficial for OOD detection. Although POT faces a performance degradation with the decrease of test batch size, its performance of lower batch sizes still hold superiority over the baseline methods. For instance, POT already achieves 93.18% average AUROC with the test batch size of 32, which outperforms the competitive rival NAC-UE (see Table 1). In Figure 4 (b), we can observe that as the regularization coefficient $\lambda$ increases, AUROC undergoes an ascending interval followed by a decrease interval, ultimately leading to convergence. In Figure 4 (c), we find that POT works better with a moderate $\omega$. This is intuitive as a lower $\omega$ may lead the generated virtual outliers close to ID samples, while a higher $\omega$ can generate virtual outliers far away both ID and OOD samples, leading the transport costs to virtual outliers indistinguishable.

**Penultimate layer vs. logit layer.** In this paper, we follow the convention in feature-based methods by using the representations from the penultimate layer of neural network, as it is believed to preserve more information than the output from the top layer, also referred to as logits. To investigate the impact of layer selection, we provide evaluation on POT using representations from the penultimate layer or logits, with the average results presented in Table 6. From the results, using the representations from the penultimate layer achieves better performance than using the logits.

Table 6: OOD detection performance comparison from penultimate and logits layer.

| Benchmarks | OOD | Logits | | Penultimate | |
|---|---|---|---|---|---|
| | | FPR95↓ | AUROC↑ | FPR95↓ | AUROC↑ |
| CIFAR-100 | Far | 18.86 | 95.58 | **14.35** | **96.70** |
| | Near | 45.68 | 85.38 | **44.28** | **86.51** |
| ImageNet-1k | Far | 20.26 | 95.49 | **15.59** | **96.78** |
| | Near | 67.97 | 80.45 | **60.15** | **83.95** |

## 5 CONCLUSION

In this paper, we tackle the problem of OOD detection from a new perspective of measuring distribution discrepancy and quantifying the individual contribution of each test input. To this end, we propose a novel method named POT, which utilizes the prototype-based OT to assess the discrepancy between test inputs and ID prototypes and use the obtaining transport costs for OOD detection. By generating virtual outliers to approximate the OOD region, we combine the transport costs to ID prototypes with the costs to virtual outliers, resulting in a more effective contrastive transport cost for identifying OOD inputs. Experimental results demonstrate that POT achieves better performance than 21 current methods. Moreover, we show that POT is pluggable with existing training methods for OOD detection and is applicable to the scenarios where the training data is unavailable, highlighting its generic nature and high practicality.

**Limitations.** Beyond extrapolation, other data augmentation techniques, such as Mixup (Zhang et al., 2018) and negative data augmentation (Sinha et al., 2021), are also employed in OOD detection

methods (Zhang et al., 2023a; Yang et al., 2023). So how to generate outliers that best fits our approach, along with the underlying theoretical analysis can be further explored. Additionally, given the generic nature of our method, combining it with other OOD detection methods is one of the potential directions for improvement.

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

# A  DERIVATION OF SINKHORN-KNOPP ALGORITHM

Recall that the classical optimal transport (OT) problem, formulated as Equation 2, is a linear programming problem that incurs cubic time complexity, which is prohibitive in many applications. To this end, we investigate a smoothed version with an entropic regularization term, formulated as Equation 4. As a $\lambda$-strongly convex function, it has a unique optimal solution. By introducing the associated Larangian, we can transform the primal problem with constraints into an unconstrained optimization problem, resulting in the following formulation:

$$\mathcal{L}(\boldsymbol{\gamma}, \boldsymbol{\mu}, \boldsymbol{\nu}) = \langle \boldsymbol{E}, \boldsymbol{\gamma} \rangle - \lambda H(\boldsymbol{\gamma}) - \boldsymbol{u}^\top (\boldsymbol{\gamma} \mathbf{1}_C - \boldsymbol{\mu}) - \boldsymbol{v}^\top (\boldsymbol{Q}^\top \mathbf{1}_m - \boldsymbol{\nu}), \tag{11}$$

where $\boldsymbol{v}$ and $\boldsymbol{v}$ are Lagrange multipliers. Taking partial derivation on transport plan $\gamma$ yields,

$$\frac{\partial \mathcal{L}(\boldsymbol{\gamma}, \boldsymbol{u}, \boldsymbol{v})}{\gamma_{ij}} = E_{ij} + \lambda \log \gamma_{ij} - u_i - v_j = 0$$

$$\gamma_{ij} = e^{(u_i - E_{ij} + v_j)/\lambda} = \boldsymbol{a}_i K_{ij} \boldsymbol{b}_j \tag{12}$$

where $\boldsymbol{a}_i = e^{u_i/\lambda}$, $\boldsymbol{b}_j = e^{v_j/\lambda}$, and $\boldsymbol{K} = \exp(-\boldsymbol{E}/\lambda)$. Conveniently, the solution can be rewritten in matrix form as $\boldsymbol{\gamma} = \text{Diag}(\boldsymbol{a})\boldsymbol{K}\text{Diag}(\boldsymbol{b})$. In this way, solving the primal problem in Equation 4 equals to finding the scaling variables $\boldsymbol{a}$ and $\boldsymbol{b}$, which must satisfy the following equations corresponding to the mass conservation constraints inherent to $\Pi(\boldsymbol{\mu}, \boldsymbol{\nu})$:

$$\text{Diag}(\boldsymbol{a})\boldsymbol{K}\text{Diag}(\boldsymbol{b})\mathbf{1}_m = \text{Diag}(\boldsymbol{a})(\boldsymbol{K}\boldsymbol{b}) = \boldsymbol{\mu}$$

$$\text{Diag}(\boldsymbol{b})\boldsymbol{K}^\top \text{Diag}(\boldsymbol{a})\mathbf{1}_C = \text{Diag}(\boldsymbol{b})(\boldsymbol{K}^\top \boldsymbol{a}) = \boldsymbol{\nu} \tag{13}$$

Known as a matrix scaling problem, the equation can be solved by modifying $\boldsymbol{a}$ and $\boldsymbol{b}$ iteratively:

$$\boldsymbol{a} \leftarrow \boldsymbol{\mu} \oslash (\boldsymbol{K}\boldsymbol{b}), \quad \boldsymbol{b} \leftarrow \boldsymbol{\nu} \oslash (\boldsymbol{K}^\top \boldsymbol{a}), \tag{14}$$

where $\oslash$ denotes element-wise division. The above iterative updates define the Sinkhorn-Knopp algorithm.

# B  HYPERPARAMETERS

POT involves two hyperparameters, i.e., the entropic regularization coefficient $\lambda$ and the linear extrapolation parameter $\omega$. To search for the optimal values, we leverage the ID and OOD validation sets in all of our experiments. Specifically, we search $\lambda$ in [1, 1.2, 1.5, 1.7, 2, 2.2, 2.5, 2.7,3], and $\omega$ in [1.5, 2, 2.5, 3] across all architectures and benchmarks.

For clarify, we provide a detailed description of validation. The validation inputs consist a mixture of samples from both ID and OOD validation sets. During validation, for each batch of validation inputs, we apply POT to calculate the transport cost scores for the samples. Based on the scores across all validation inputs, we calculate the evaluation metrics (e.g., AUROC or FPR95) to select the optimal hyperparameters $\lambda$ and $\omega$ that achieve the best performance for POT.

# C  FULL EXPERIMENTAL RESULTS ON IMAGENET

| Method | Vit-b16 | | | | ResNet-50 | | | |
|---|---|---|---|---|---|---|---|---|
| | iNaturalist | OpenImage-O | Textures | Average | iNaturalist | OpenImage-O | Textures | Average |
| OpenMax | 94.91 | 87.37 | 85.54 | 89.27 | 92.03 | 87.59 | 88.13 | 89.25 |
| MSP | 88.16 | 84.85 | 85.09 | 86.03 | 88.43 | 84.99 | 82.48 | 85.30 |
| ODIN | 79.53 | 71.46 | 77.24 | 76.08 | 91.13 | 88.27 | 89.01 | 89.47 |
| MDS | 96.00 | 92.35 | 89.39 | 92.58 | 63.67 | 68.74 | 89.72 | 74.04 |
| MDSEns | / | / | / | / | 61.95 | 61.14 | 80.02 | 67.71 |
| RMDS | 96.09 | 92.29 | 89.38 | 92.59 | 87.27 | 85.73 | 86.07 | 86.36 |
| EBO | 79.26 | 76.49 | 81.22 | 78.99 | 90.61 | 89.15 | 88.74 | 89.50 |
| GradNorm | 42.36 | 37.83 | 45.05 | 41.75 | 93.85 | 85.11 | 92.08 | 90.35 |
| ReAct | 86.09 | 84.26 | 86.69 | 85.68 | 96.32 | 91.88 | 92.80 | 93.67 |
| MLS | 85.25 | 81.61 | 83.79 | 83.55 | 91.15 | 89.26 | 88.42 | 89.61 |
| KLM | 89.54 | 86.96 | 86.51 | 87.67 | 90.77 | 87.35 | 84.69 | 87.60 |
| VIM | 95.71 | 92.15 | 90.61 | 92.82 | 89.54 | 90.40 | **97.96** | 92.63 |
| KNN | 91.45 | 89.83 | 91.12 | 90.80 | 86.31 | 86.78 | 97.06 | 90.05 |
| DICE | 82.55 | 82.33 | 82.26 | 82.38 | 92.50 | 88.46 | 92.07 | 91.01 |
| RankFeat | / | / | / | / | 40.10 | 50.94 | 70.96 | 54.00 |
| ASH | 50.47 | 55.45 | 47.87 | 51.26 | 97.04 | 93.31 | 96.91 | 95.76 |
| SHE | 93.56 | 91.03 | 92.67 | 92.42 | 92.58 | 86.70 | 93.63 | 90.97 |
| GEN | 93.53 | 90.25 | 90.25 | 91.34 | 92.44 | 89.35 | 87.63 | 89.81 |
| DDE | 97.10 | 88.97 | 88.96 | 91.68 | 83.64 | 71.38 | 86.84 | 80.62 |
| NAC-UE | 93.69 | 91.54 | 94.17 | 93.13 | 96.43 | 91.61 | 97.77 | 95.27 |
| POT | **99.38** | **95.60** | **95.36** | **96.78** | **99.45** | **93.45** | 95.58 | **96.16** |

Table 7: Far-OOD detection performance on ImageNet-1k benchmark. We report the AUROC↑ scores over two backbones, i.e., ResNet-50 and Vit-b16).

| Method | Vit-b16 | | | ResNet-50 | | |
|---|---|---|---|---|---|---|
| | SSB-hard | NINCO | Average | SSB-hard | NINCO | Average |
| OpenMax | 68.81 | 78.68 | 73.74 | 71.30 | 78.12 | 74.71 |
| MSP | 69.03 | 78.09 | 73.56 | 72.16 | 79.98 | 76.07 |
| ODIN | 63.69 | 65.20 | 64.45 | 71.84 | 77.73 | 74.78 |
| MDS | 71.45 | 86.49 | 78.97 | 47.15 | 62.19 | 54.67 |
| MDSEns | / | / | / | 44.89 | 55.64 | 50.27 |
| RMDS | 72.79 | 87.28 | 80.04 | 71.47 | 82.22 | 76.85 |
| EBO | 59.16 | 66.05 | 62.60 | 72.35 | 79.70 | 76.03 |
| GradNorm | 43.27 | 35.63 | 39.45 | 72.83 | 73.93 | 73.38 |
| ReAct | 63.24 | 75.45 | 69.34 | 73.09 | 81.71 | 77.40 |
| MLS | 64.45 | 72.39 | 68.42 | 72.75 | 80.41 | 76.58 |
| KLM | 68.11 | 80.64 | 74.37 | 71.19 | 81.87 | 76.53 |
| VIM | 69.34 | 84.63 | 76.99 | 65.10 | 78.54 | 71.82 |
| KNN | 65.93 | 82.22 | 74.07 | 61.78 | 79.41 | 70.60 |
| DICE | 60.01 | 71.91 | 65.96 | 70.84 | 75.98 | 73.41 |
| RankFeat | / | / | / | 55.87 | 46.10 | 50.98 |
| ASH | 54.12 | 53.07 | 53.60 | 73.11 | **83.37** | 78.24 |
| SHE | 68.11 | 84.16 | 76.13 | 71.83 | 76.42 | 74.13 |
| GEN | 70.19 | 82.47 | 76.33 | 72.11 | 81.71 | 76.91 |
| DDE | 76.29 | 81.32 | 78.81 | 56.52 | 62.93 | 59.72 |
| NAC | 68.04 | 82.45 | 75.25 | 68.21 | 81.16 | 74.68 |
| POT | **80.39** | **87.51** | **83.95** | **83.37** | 78.13 | **80.75** |

Table 8: Near-OOD detection performance on ImageNet-1k benchmark. We report the AUROC↑ scores over two backbones, i.e., ResNet-50 and Vit-b16.

