# OpenReview forum: "Prototype-based Optimal Transport for Out-of-Distribution Detection"
_ICLR.cc/2025/Conference — Submitted to ICLR 2025_

### Official Review · Reviewer_zWXM · 2024-10-26

**Soundness:** 1
**Presentation:** 2
**Contribution:** 2
**Rating:** 3
**Confidence:** 5

**Summary:**

This paper introduces an approach to out-of-distribution (OOD) detection leveraging optimal transport (OT). The core idea is based on the observation that OOD inputs typically incur higher transport costs when mapped to the in-distribution (ID) in latent space. From this, they develop an OOD detection method by first compute the optimal transport plan from the test distribution to the ID distribution. OOD detection is then achieved by calculating the transport cost for each input and applying a threshold to determine whether the input is OOD. The author then benchmark their method with varied number of datasets.

**Strengths:**

- The comparison with other baseline methods appears comprehensive and thorough.

**Weaknesses:**

One of my concern is that  the paper lacks a deeper understanding for why OT would be better than other OOD detection methods. Standard approaches have increasingly focused on understanding geometric representations in neural networks, either through uncertainty estimation [1] or, more rigorously, likelihood estimation [2], sometimes aided by reconstruction autoencoders [3]. Given the connection between geometric representation and OT through the transport map, I would expect interpretative insights along this line rather than a direct application of OT for OOD detection. Without this, the current proposal, in my opinion, is incremental compared to the existing distance-based approaches.

The method presentation is not clear. I have some confusion in terms of how the OT plan is constructed using both in-distribution (ID) inputs and out-of-distribution (OOD) datasets. See question (1).

Finally, similar ideas of using optimal transport to characterize OOD error have been explored with more concrete theoretically properties, as in [4].

[1] Cao, Senqi, and Zhongfei Zhang. "Deep hybrid models for out-of-distribution detection." Proceedings of the IEEE/CVF Conference on Computer Vision and Pattern Recognition. 2022.

[2] Kamkari, Hamidreza, et al. "A Geometric Explanation of the Likelihood OOD Detection Paradox." Forty-first International Conference on Machine Learning.

[3] Zhou, Yibo. "Rethinking reconstruction autoencoder-based out-of-distribution detection." Proceedings of the IEEE/CVF Conference on Computer Vision and Pattern Recognition. 2022.

[4] Lu, Yuzhe, et al. 'Characterizing out-of-distribution error via optimal transport.' Advances in Neural Information Processing Systems 36 (2023): 17602-17622."

**Questions:**

(1) Can the author specify how the validation set is built in the OT formulation, especially the test batch size in Section 4.3? Specifically, my impression is that we only have 1 unique transport map from the validation set and we will use that for detecting OOD samples. As such, I'm not sure how varying the test batch size affect the result. Are we computing a distinct transport map for every test batch size? if so how do we decide the threshold for OOD samples for these test batch size? What if their statistic (portion between ID and OOD data) changes?

(2) How to apply this idea when the OOD set is small? How the validation set size affect the performance?

(3) What is the fundamental idea that OT is better than other methods such as likelihood estimation?

---

> ### Author Response · Authors · 2024-11-21
> **Thank you; new experiments and justifications [1/2]**
>
> **Q1:** Can the author specify how the validation set is built in the OT formulation, especially the test batch size in Section 4.3? I'm not sure how varying the test batch size affect the result. Are we computing a distinct transport map for every test batch size? if so how do we decide the threshold for OOD samples for these test batch size? What if their statistic (portion between ID and OOD data) changes?
>
> **A1:** Thank you for raising this concern. We will answer the question with the following three points:
>
> 1. **The different functions of validation sets.** We would like to clarify that, unlike the validation set built from the training data to learn a threshold in the problem of OOD performance prediction (e.g., [1]), the validation sets in OOD detection are used to select hyperparameters before the test phase.
> That is, in OOD detection, the validation sets are built from other datasets in advance as described in line 298-303.
> By the way, we do not need to decide specific thresholds in the experiments since the threshold for determining whether a sample is OOD depends on the requirements of real-world applications, such as the score making 5\% ID samples being falsely identified as OOD. In literature, the used evaluation metrics (i.e., AUROC and FPR95 ) are threshold-free.
>
> 2. **The calculation of OT plans.** For each test batch, our method computes a distinct optimal transport map between ID prototypes and the test batch, deriving a score for each test sample. The evaluation metrics are computed over all test samples’ scores.
> The reason why test batch size affects the result is that a larger batch result in the empirical distribution (i.e., the observed data distribution) better approximating the real test data distribution, enabling a more accurate measurement of distribution differences and better performance.
>
> 3. **The robustness to the changeable ID/OOD ratios.** In the experiments, there are actually changes in ID/OOD ratios: (1) the numbers of OOD samples in different OOD datasets are quiet different ranging from 5160 (Textures) to 49000 (SSB-hard) with the same number of test ID samples, resulting in different ID/OOD ratios across different datasets; (2) the ID/OOD ratios also vary across different batches in the same dataset due to random batch division. But our method has consistently achieved superior results and outperformed its competitors, demonstrating the robustness to the changeable ID/OOD ratios.
>
> ---
>
> **Q2:** How to apply this idea when the OOD set is small? How the validation set size affect the performance?
>
> **A2:** 1. Although the performance of POT slightly declines with the decrease of test size, it still outperforms the baselines with small test size. As an evidence, we conducted experiments on the ImageNet benchmark with the test size of 64, and the results are shown below:
>
> | Datasets          | VIM   | KNN   | ASH   | SHE   | GEN   | DDE   | NAC-UE | **POT**   |
> |--------------------|-------|-------|-------|-------|-------|-------|--------|-----------|
> | iNaturalist       | 95.71 | 91.45 | 50.47 | 93.56 | 93.53 | 97.10 | 93.69  | **99.04** |
> | Openimage-O       | 92.15 | 89.83 | 55.45 | 91.03 | 90.25 | 88.97 | 91.54  | **93.94** |
> | Textures          | 90.61 | 91.12 | 47.87 | 92.67 | 90.25 | 88.96 | **94.17** | 92.34     |
> | **Average (Far-OOD)** | 92.82 | 90.80 | 51.26 | 92.42 | 91.34 | 91.68 | 93.13  | **95.11** |
> | SSB-hard          | 69.34 | 65.93 | 54.12 | 68.11 | 70.19 | 76.29 | 68.04  | **77.31** |
> | NINCO             | **84.63** | 82.22 | 53.07 | 84.16 | 82.47 | 81.32 | 82.45  | 83.10     |
> | **Average (Near-OOD)** | 76.98 | 74.08 | 53.59 | 76.13 | 76.33 | 78.81 | 75.25  | **80.20** |
>
> The results indicate our method already outperforms existing methods even with a test set size of only 64. Therefore, we believe our method can be applied with no problem to small OOD set.
>
> 2. The validation set is used to select hyperparameters before test phase as clarified above, and its size has little affect to the performance.

---

> ### Author Response · Authors · 2024-11-21
> **Thank you; new experiments and justifications [2/2]**
>
> **Q3:** What is the fundamental idea that OT is better than other methods such as likelihood estimation?
>
> **A3:** The common idea behind many post-hoc methods, including likelihood estimation approaches, is to recognize the known distribution of ID data by learning their commonality or density function.
> Samples that deviate from the known ID distribution, either by departing from the commonalities or falling into low-likelihood regions, are then identified as OOD.
> However, these methods focus exclusively on the ID distribution and overlook the underlying OOD distribution.
>
> In contrast, our method overcomes this limitation by considering not only the ID distribution but also the underlying OOD distribution itself.
> Recall the motivation of our method that the distribution discrepancy between test inputs (i.e., a mixture of ID and OOD data) and pure ID data is primarily caused by the presence of OOD data, which can identified by quantifying each sample's contribution to the overall discrepancy.
> By implicitly exploring and utilizing the information about the OOD distribution embedded in the test inputs, our method identifies OOD samples more effectively, accounting for both ID and OOD data distributions.
>
> On the other hand, as a tool for measuring distribution discrepancy, OT offers several appealing advantages for OOD detection. First, OT enables a fine-grained assessment of individual sample contributions to the overall discrepancy, making the transport cost a critical indicator for identifying OOD samples. Second, the transport cost captures the geometric differences between ID and OOD representations in the latent space, providing a geometrically meaningful interpretation.
>
> Finally, our comprehensive experiments have demonstrated the superior performance of our method over existing detection approaches.
>
> ---
> **Q4:** Similar ideas of using optimal transport to characterize OOD error have been explored with more concrete theoretically properties, as in [1].
>
> **A4:** Our method fundamentally differs from existing methods using OT to characterize OOD error (e.g., [1]) in the following key aspects:
>
> 1. **Different problem settings.** OOD Detection aims to identify individual OOD samples in test data, while [1] addresses the estimation of the overall OOD error for a test set. Importantly, [1] assumes that the label distribution of test data is close to that of the training data.
> This assumption does not hold in OOD detection, where OOD samples inherently belong to unknown classes with labels distinct from the training data.
>
> 2. **Different OT modeling and utilization.**
> Due to the different problem settings and goals, the application of OT in our method and in [1] diverges significantly.
> In [1], OT is employed to estimate the difference between the predicted and true OOD label distributions by calculating transport plan between the distribution of confidence vectors of test data and the label distribution of training data. We need to emphasize that this approach relies on the assumption of identical label distributions between training and test data, which is invalid in OOD detection.
> In contrast, our method leverages OT between the representations of test data and ID prototypes to measure the distribution difference in the latent space and quantify each test sample's contribution to indicate its likelihood of being an OOD sample.
> The behind motivation is straightforward: ID and OOD data inherently exhibit distribution difference and thus OOD samples contribute more to the difference between ID and test data.
> In other words, apart from the difference in the measured space (i.e., label space v.s. latent space), [1] focuses on assessing the overall difference while our method further utilizes OT to derive fine-grained sample-wise contributions, which are necessary for OOD detection.
>
> We would like to highlight that our paper centers on proposing an intuitive perspective of measuring distribution difference for OOD detection, and OT serves as a powerful and suitable tool for achieving this goal.
> Meanwhile, to the best of our knowledge, our method is the first post-hoc OOD detection approach applying OT.
>
> [1] Lu, Yuzhe, et al. 'Characterizing out-of-distribution error via optimal transport.' Advances in Neural Information Processing Systems 36 (2023): 17602-17622."

---

> ### Comment · Reviewer_zWXM · 2024-11-21
> **Further Question (1/2)**
>
> Thank you for your response, can you help me clarify the following points?
>
>
> (A1) **The different functions of validation sets.**: so which hyperparameters are used in the paper? Is it the $\lambda$ in Equation (4)? This information needs to be highlighted in the paper.
>
> **The calculation of OT plans**: So given a test batch (let's say it has a size of 1000 data), we construct an OT map from this test batch and to the ID prototypes. Then, we compute the score for each test point from the OT map. Then, we decide which test points within these 1000 data points are OOD based on this OT plan. Is it a correct interpretation? Can you also give us more details how the test batch is constructed? Let's say my ID distribution is MNIST, my validation OOD is CIFAR 10 and my OOD test dataset is Imagenet.
>
> I'll ask questions on other points after we clarify this. Thank you for your time. Overall my impression is that this paper needs significant adjustment since none of these important points are described in the paper.

---

> > ### Author Response · Authors · 2024-11-23
> > **Thank you for your prompt reply**
> >
> > 1. The hyperparameters used in the paper is λ in Equation (4) and ω in Equation (9). According to the ablation study on the two hyperparameters in Section 4.3 (Figures 4 (b)&(c)), the performance differences under varying hyperparameter settings are not significant (less than 10 percent in AUROC), demonstrating that our method is not sensitive to hyperparameter choices. Additionally, our method requires fewer hyperparameters compared to NAC, which has four, and the optimal hyperparameter ranges are similar across different OOD datasets. This suggests the ease of hyperparameter selection and highlights the practical utility of our method in real-world applications.
> >
> > 2. We appreciate your careful reading, and your understanding of the OT plan calculation is correct.
> > To clarify further, we outline the detailed process based on the settings you provided:
> > * Assuming ID dataset is MNIST consisting of a training set and a test set, the OOD validation dataset is CIFAR 10 and the OOD test dataset is Imagenet, **validation inputs** are a mixture of MNIST test set and CIFAR 10, and **test inputs** are a mixture of MNIST (ID) test set and Imagenet (OOD) test set. The ID prototypes are constructed from the MNIST training set.
> > * During the **validation phase**, for each batch of validation inputs, we construct an OT map between this batch and the ID prototypes to compute the scores for the samples.
> > After aggregating the scores across all the validation samples, we calculate the evaluation metrics (e.g., AUROC or FPR95) to select the optimal hyperparameters λ and ω that achieve the best performance for our method.
> >
> > * During the **test phase**, we evaluate POT over the test inputs with the selected λ and ω using validation sets.
> > The process is similar: for each test batch, we construct an OT map between the batch and the ID prototypes to compute the scores. After aggregating the scores across all test samples, we calculate the evaluation metrics for comparison.
> >
> > Unlike most existing methods, our method does not involve learning a density function for the ID data or training a detector prior to encountering a test set. Instead, it dynamically measures the distribution discrepancy between a test batch and the ID prototypes on the fly.
> > This design makes our method highly easy-to-use, offering an effective solution while facilitating rapid deployment in real-world production environments.
> >
> > We sincerely apologize for the confusion and have revised our paper for greater clarity. The updates are **highlighted in red** and are summarized as follows:
> >
> > * **Lines 304-306** explicitly specify the hyperparameters used in POT, with detailed tuning process described in **Appendix B**.
> >
> > * **Lines 309-315** provide additional details to clarify the test phase.

---

> ### Comment · Reviewer_zWXM · 2024-11-23
> **Thank you, further question**
>
> Thank you for your reply. I have further questions. So during the **test phase**, how should I choose the threshold to detect OOD samples?
>
> Since there are only a few days left until the rebuttal ends, I will write down my concerns and hope the authors can follow up. My concern is that this method is very dependent on the test distribution. Let's say my ID prototype is a 0  from the ID distribution $P_X$ which is a $\mathcal{N}(0,0.01)$ and I have two test distributions, says normal distribution $\mathcal{N}(0,0.01)$ (call $P_A = P_X$, same distribution) and $\mathcal{N}(10.0,1.0)$ (call $P_B$).
>
> With distribution $P_A$, I set the interval $[-a, a]$ as a threshold to satisfy FPR@95. For test distribution $P_B$, similarly, I have interval $[-b, b]$, where $b \neq a$. My impression is that there are points that lie within $[-b, -a]$ and $[a,b]$ that are sometimes classified as OOD and sometimes not. This is not standard in OOD detection where we would like to point out which data point is considered OOD to the original input distribution. Let me know if I misunderstood anything and I'll happily adjust my score accordingly.
>
> On the fundamental idea difference between OT and likelihood estimation, I still do not see the advantage since there is no rigorous justification. It would be better if the author provide a toy example, 2D data is totally good for me, to demonstrate the geometric advantage offered by OT that likelihood estimation fails to.

---

> ### Author Response · Authors · 2024-11-24
> **Thank you for the reply**
>
> **Response to the threshold selection:**
> As mentioned in **A1 1.** to you, we do not need to decide a specific threshold during the test phase as the widely used metrics (i.e., AUROC and FPR95) are **threshold-free**. For the practice, a commonly recommended threshold in most OOD detection works is the score at which 5% ID samples being falsely identified as OOD (corresponding to FPR95).
> The threshold, akin to a hyperparameter, can be determined using an ID validation set.
> Specifically, we can apply POT between the ID prototypes and the ID validation set to calculate scores for the ID samples. The score of the 95th percentile of the validation samples is selected as the threshold.
>
> We try to understand that your concern involves that the threshold for test distribution A may not achieve the same performance for test distribution B (e.g. satisfying FPR95).
> To mitigate the influence of test distribution on threshold, we can incorporate some ID validation samples into the test inputs, and also use the 95th percentile score of the validation samples as threshold.
> Since the validation samples share the same distribution as the original ID distribution, the resulting threshold will still ensure satisfying FPR95 even the test distribution varies.
>
> We would like to clarify that almost all the OOD detection studies only utilize the threshold-free evaluation metrics as they give a holistic measure of performance.
> The above discussion comes from our familiarity about the OOD detection field and POT, and does not affect the academic contribution of this paper.
>
> ---
>
> **Response to the method comparison:**
> Since the mechanisms and output results of our method and likelihood estimation are quite different, it is challenging to provide a toy example. Instead, I highlight their differences from the follow perspective:
>
> 1. Likelihood estimation generates a data distribution to approximate the ID distribution, with its OOD detection performance heavily dependent on the quality of the generated distribution, which can be more pronounced for complex ID distributions.
> In contrast, OT does not involve data generation or learning the density function of the ID distribution. Instead, it dynamically measures the distribution discrepancy between test inputs and ID data on the fly.
> In other words, OT actually does not focus on the specific ID distribution and bypasses the need to approximate or learn it. This eliminates the requirement for complex preprocessing and avoids the biases that may be introduced through generated data distributions or learned density functions, ensuring more robust OOD detection.
> Additionally, likelihood estimation relies on specific density models, such as diffusion models, while our method is model-agnostic and can be applied to different off-the-shelf models.
>
> 2. Likelihood estimation focuses solely on modeling the ID distribution and may assign low confidence to ID samples located near the boundaries of the ID distribution, potentially misclassifying them as OOD.
> In contrast, OT takes into account the overall geometry of the sample space, including the OOD samples present in the test inputs.
> Specifically, OT prioritizes the mass transport of ID samples, including those at the distribution edges, over the OOD samples that are farther from the ID data.
> The global consideration between different distributions allows our method to effectively capture geometric differences between ID and OOD samples through transport costs, leading to distinct separation of them.
>
> We sincerely appreciate your insights and questions. Please feel free to have further discussion with us.

---

> ### Comment · Reviewer_zWXM · 2024-11-24
>
> "To mitigate the influence of test distribution on the threshold, we can...", this is partially my concern, deciding whether a data point is OOD or not should not depend on how the test distribution comes, e.g. an MNIST image that accidentally ends up in ImageNet dataset should still be classified as ID if the ID distribution is MNIST. As far as I know, all OOD detection methods assign a single score for any data point.
>
> In other words, a data point $x$ belonging to the test batch that follows the distribution $P_A$ will have a different transport cost than the same data point $x$ but now in another test batch that follows the test distribution $P_B$, since their transport plans are constructed adaptively. Hence, the score of $x$ is decided by which distribution we are testing. For this reason, there will be a threshold where $x$ is OOD when it's coming from $P_A$ but not from $P_B$ (or vice versa). I'm aware that the evaluation metric is threshold-free, nevertheless, the problem is that the score assigned to a data point in this method is not invariant w.r.t the test distribution, which is why I was asking in detail about the thresholding. I now share the same concern as Reviewer 7P4j on this matter of using test distribution to compute the OOD score, which I previously thought to be obtained from the validation dataset. Please feel free to correct me if my understanding is not correct.

---

> > ### Author Response · Authors · 2024-11-25
> > **Thank you for the reply**
> >
> > We understand your concern that the score of the same sample may vary in different batches and could be identified as ID or OOD w.r.t a given threshold. However, as mentioned in our previous reply, we can address this by inserting some known ID samples into the test inputs, and using the 95th percentile score of the these ID samples as threshold.
> > **While the threshold may also vary across different batches, it ensures that the samples in each batch are identified to satisfy FPR95, as these ID samples share the same distribution as the original ID distribution.**
> > Moreover, as demonstrated in our response **A2 to Reviewer q2aw**, the calculation time of OT is minimal, so adding some ID samples into the test inputs only slightly increases the execution time.
> >
> > Additionally, I would like to point out that, during the test phase, the final metrics are calculated by aggregating scores across different batches to ensure a fair comparison with other methods.
> > The experimental results demonstrate that our method achieves a better mean result with very small standard deviation across multiple runs, which can support the fact that a test instance deemed as ID in bacth A will be unlikely classified OOD in batch B.
> >
> > We would be very grateful if you can re-evaluate our paper according to our discussions. Please let us know if you have any questions and we will try our best to quickly reply to you before the rebuttal deadline.

---

> ### Comment · Reviewer_zWXM · 2024-11-25
>
> Thank you for your response. However, the issue is fundamental and requires extensive discussion in the paper. For this setup, technical analysis is necessary and empirical evaluation alone is not sufficient. A variability in the OOD score for a sample across different test distributions is problematic for safety-critical AI applications. Such variability can lead to misinterpretations by practitioners. Therefore, I will keep my score.

---

> ### Author Response · Authors · 2024-11-25
> **Thank you for your reply and active discussion**
>
> We sincerely appreciate your time and effort in reviewing our work and actively engaging in discussions with us.
> However, we still would like to clarify that not all OOD detection methods assign identical scores to a given sample. For example, DDE [R1] and RTL [R2] may also produce varying scores for the same sample in different test distributions. This score variability is a common phenomenon in methods that utilize test distributions, but we firmly believe it does not diminish their significant practical value for the following reason:
>
> 1. As we discussed above, OOD scores work in conjunction with thresholds to identify OOD samples. It is the threshold choice that matters in real-world applications. Almost all existing OOD methods advocate for selecting thresholds with the 95th percentile of ID validation samples to satisfy the FPR95 metric. In the last reply, we indeed provided a practicable approach to determine such thresholds using ID validation samples, ensuring the required metrics like FPR95 are satisfied across different test batches. This aligns with the advocated practices and demonstrates that our method can be reliably applied in real-world applications.
>
> 2. On the other hand, in most OOD detection methods, the OOD score of a sample does not represent a practical quantity, such as the true probability of being OOD. In other words, the absolute value of an OOD score is not useful. While other methods without score variability may allow direct comparisons of OOD likelihood between samples via OOD scores, our method can achieve the same comparisons by incorporating the samples to be compared within the same input batch and calculating their scores accordingly.
>
> Therefore, in our view, score variability is not a big issue whether in the academic evaluation of the method performance or in the practical real-world applications. And applying our method reasonably will not lead to misinterpretations for practitioners. If you have any further questions, we are very welcome to discuss with you.
>
> [R1] In- or Out-of-Distribution Detection via Dual Divergence Estimation. UAI. 2023.\
> [R2] Test-Time Linear Out-of-Distribution Detection. CVPR. 2024.

---

### Official Review · Reviewer_7P4j · 2024-11-01

**Soundness:** 3
**Presentation:** 2
**Contribution:** 2
**Rating:** 3
**Confidence:** 5

**Summary:**

This paper addresses the inherent distributional differences between ID and OOD samples by proposing the use of optimal transport (OT) to measure the discrepancy between test inputs and ID prototypes, a method termed Prototype-based Optimal Transport (POT). The authors assess distances by calculating the transport cost from each test input to the ID prototypes. Additionally, they apply linear extrapolation to generate virtual outliers, effectively regularizing the decision boundary. By evaluating distributional differences between ID and OOD samples through optimal transport, this work introduces a novel perspective. Furthermore, the use of linear extrapolation enables optimal transport to function effectively within this framework.

**Strengths:**

1.	Explore a new OOD score by measuring the discrepancy between test images and ID prototypes from the perspective of optimal transport, which is well-supported theoretically.
2.	Use linear extrapolation to regularize the decision boundary, offering a successful perspective on applying optimal transport in OOD detection.

**Weaknesses:**

1.	In the use of OT, calculating ν for test samples in Equation (3) requires all test samples, which is impractical in real-world applications.
2.	When generating outliers via linear extrapolation, the method appears to rely on information from the test set, which is also impractical.

**Questions:**

Why do you claim that your method measures differences directly in the latent space? It is clear that you are also using features obtained from the feature extractor. When comparing with DDE, you state that DDE measures differences in the representation space, which is influenced by the transformations of the DNN features, while POT measures differences in the latent space.

I will adjust the score based on your response.

---

> ### Author Response · Authors · 2024-11-21
> **Thank you; justifications**
>
> **Q1:** In the use of OT, calculating v for test samples in Equation (3) requires all test samples, which is impractical in real-world applications.
>
> **A1:** Thank you for highlighting this concern. Actually, our method is designed under the assumption that test data arrive in batches, rather than requiring access to the entire test dataset simultaneously.
>
> Batch-based inputs are a common and realistic setting in numerous real-world applications. For instance: (1) In industrial settings, sensor data from manufacturing lines are typically analyzed in batches to detect anomalies; (2) In healthcare applications, patient data often arrives in groups (e.g., daily or hourly reports) for processing and analysis.
>
> Additionally, in some related topics, such as continual learning, it is typical to encounter OOD inputs from unseen classes due to domain nonstationarity.
> It is required to identify the OOD samples in test inputs and extend the classification ability to them. In such cases, batched test inputs are typically collected for network retraining.
>
> By leveraging batch-based inputs, our method aligns with the typical data processing workflows in real-world scenarios. This design ensures both practicality and scalability, making our approach suitable for deployment across a wide range of applications.
> We will add more discussions on the rational of batch-based test inputs for clarity in the revised manuscript.
>
> ---
>
> **Q2:** When generating outliers via linear extrapolation, the method appears to rely on information from the test set, which is also impractical.
>
> **A2:** Thank you for raising this concern. We would like to argue that our solution is both practical and feasible in real-world applications, for the following reasons:
>
> 1. Our method does not require any label information about test samples.
>
> 2. Our method only utilizes the empirical mean of test inputs to help generate virtual outliers, and such information can be easily collected in the test phase.
>
> 3. Utilizing the empirical mean of test inputs is a common practice in data processing.
> For instance, empirical mean and variance of batch data are used in batch normalization layers of DNNs to mitigate internal covariate shift.
>
> ---
>
> **Q3:** Why do you claim that your method measures differences directly in the latent space? It is clear that you are also using features obtained from the feature extractor. When comparing with DDE, you state that DDE measures differences in the representation space, which is influenced by the transformations of the DNN features, while POT measures differences in the latent space.
>
> **A3:** The sample features in the latent space refer to the outputs from the hidden layers of models.
> Our method directly utilizes these latent space features, specifically the features obtained from the feature extractor, to measure distributional differences. In contrast, DDE estimates dual KL-Divergence (KL-D) by transforming these latent space features into a dual space using a DNN, i.e., DNN features.
> This approach is adopted because estimating the KL-D of latent space features is challenging due to the unknown distribution of test data.
>
> However, such transformation introduces several limitations for DDE: (1) training the DNN for transformation requires optimizing a non-convergent loss function and handling multiple parameters. This makes the separability of ID and OOD features uncertain, potentially leading to indistinguishability between ID and OOD samples or unintended separations within ID samples; (2) the dual KL-D is calculated using features transformed by the DNN, where the transformation process acts as a black box. This can lead to information loss from the original features, resulting in an inaccurate estimation for the true distribution differences between ID and OOD data.
>
> In contrast, our method avoids these drawbacks by directly measuring differences in latent space features using optimal transport. Since OOD data are not involved in the training process, the latent space features naturally encode information about the distribution shift between ID and OOD data. This enables our method to accurately capture the original distribution differences without the need for training additional DNNs, ensuring reliable and robust OOD detection.

---

> > ### Author Response · Authors · 2024-11-24
> > **Thank you; invitation to reply**
> >
> > Dear reviewer,
> >
> > We truly appreciate your time and effort in reviewing our work. We hope our response has addressed your concerns. Please do not hesitate to let us know if you have any further questions or feedback.

---

### Official Review · Reviewer_jBPb · 2024-11-02

**Soundness:** 3
**Presentation:** 3
**Contribution:** 3
**Rating:** 6
**Confidence:** 3

**Summary:**

The work considered the problem of OOD (Out-Of-Distribution) detection, and proposed a method based on optimal transport from test inputs to ID (in-distribution) prototypes. It first constructed the prototype of each class as the average representation for that class extracted from the feature encoder, then computed the optimal transport between the encoded feature of the test inputs and the class prototypes and used the sum of the transport costs from a test input to prototypes as an indication of the input's outlier-ness. It then noted that only using this is insufficient for detecting OOD data closer to ID, and proposed to generate virtual outliers by linear extrapolation between the prototypes and the mean of the test inputs, and finally used the difference between the transport cost to the ID prototypes and the cost to the virtual outliers as the final OOD score.

The work then performed extensive experiments. The results showed that the proposed method often outperforms the competitors and gets close-to-best performance otherwise. It can be used with different pretraining schemes without graceful performance degradation, and can use the network weights as the ID prototypes when training data are not available. Ablation study shows that the virtual outliers are important for the performance, and that the method is robust to hyperparameters: the extrapolation coefficient, the regularization coefficient, and the test batch size.

**Strengths:**

- The work addressed an important topic of OOD detection, and designed a pratical method.
- The proposed method used two key techniques: optimal transport and virtual outliers via linear extrapolation. The work illustrate the insights behind these techniques quite clearly.
- The experiments are quite extensive and the results are quite convincing, showing the strong performance and investigating important aspects of the method like robustness to hyperparameters.

**Weaknesses:**

- See the questions below.

**Questions:**

- The performance will depend on the feature encoder used. Is the method robust to the choice of the feature encoder? How large is the impact of the encoder?
- What if we use more than one class prototypes for each class?
- What are the \mathcal{P} and \mathcal{M} in Line 241? Better to formally specify them.
- Eqn (9): the second "+" should be ":"?
- Table 1, Textures FPR95, NAC-UE result: should be underlined rather than boldfaced.
- Maybe also underline the second best results in Table 2, so as to show the performance margin and be consistent with Table 1.
- Figure 3 shows the ablation result on virtual outliers on Far-OOD datasets. What about ablation study on Near-OOD datasets?

---

> ### Author Response · Authors · 2024-11-21
> **Thank you; new experiments and explanations [1/2]**
>
> **Q1:** The performance will depend on the feature encoder used. Is the method robust to the choice of the feature encoder? How large is the impact of the encoder?
>
> **A1:** Yes, our method is robust to the choice of feature encoder.
> We have demonstrated this in Table 3, which shows the robustness of our method to different backbones, and in Table 4, which evaluates the performance with different training schemes for the feature encoder. Across these settings, our method consistently achieves excellent and stable performance, outperforming all competitors.
> The choice of encoder does not have large impact on our method. We posit that this is because, unlike other OOD detection methods, which often rely on specific attributes or patterns in the extracted ID features that can vary significantly with the encoder, our method focuses on capturing the differences between ID and OOD data. Since OOD data are never encountered during training, the distribution differences between ID and OOD features remain consistent regardless of the encoder, ensuring that our method is minimally affected by the choice of encoder.
>
> ---
>
> **Q2:** What if we use more than one class prototypes for each class?
>
> **A2:** Thank you for this suggestion. Inspired by your question, we conducted a series of experiments on the ImageNet benchmark, varying the number of prototypes per class from 1 to 10. Each prototype was constructed as the average feature representation of 50 samples from the same class. The results are as follows, which shows that the performance remains stable:
>
> |  | iNaturalist | | Textures| | OpenImage-O |  | SSB-hard |  | NINCO | |
> |------------------------|--------------------|--------------------|-----------------|-----------------|--------------------|--------------------|----------------|----------------|--------------|--------------|
> |Number for each class |  FPR95↓ |  AUROC↑ | FPR95↓ |  AUROC↑ | FPR95↓ |  AUROC↑ | FPR95↓ |  AUROC↑ | FPR95↓ |  AUROC↑ |
> | 1                      | 0.63              | 99.39             | 25.55          | 95.34          | 20.93              | 95.58             | 75.00         | 80.36         | 46.00       | 87.52       |
> | 2                      | 0.65              | 99.39             | 24.98          | 95.37          | 20.89              | 95.60             | 75.05         | 80.39         | 45.59       | 87.50       |
> | 5                      | 0.66              | 99.38             | 25.05          | 95.37          | 21.02              | 95.59             | 75.09         | 80.36         | 45.23       | 87.50       |
> | 10                     | 0.66              | 99.38             | 24.96          | 95.37          | 20.99              | 95.59             | 75.11         | 80.36         | 45.63       | 87.48       |
>
> We conjecture this is because the differences between inter-class prototypes are significantly larger than those between intra-class prototypes. Consequently, increasing the number of intra-class prototypes is analogous to duplicating the existing prototypes, which does not substantially alter the original transport plan or detection performance.

---

> > ### Author Response · Authors · 2024-11-21
> > **Thank you; new experiments and explanations [2/2]**
> >
> > **Q3:** What are the $P$ and $M$ in Line 241? Better to formally specify them.
> >
> > **A3:** We apologize for the confusion. The $P$ and $M$ refer to the prototypes and the average representation of the test inputs, respectively. We will formally define these terms in the revised paper for clarity.
> >
> > ---
> >
> > **Q4:** First, Eqn (9): the second "+" should be ":"?   Second, Table 1, Textures FPR95, NAC-UE result: should be underlined rather than boldfaced.  Third, maybe also underline the second-best results in Table 2, so as to show the performance margin and be consistent with Table 1.
> >
> > **A4:**
> > Thank you for identifying these issues:
> > 1. For Eqn (9), the second "+" will be corrected to ":" as suggested.
> > 2. In Table 1, the FPR95 of Textures for NAC-UE will be underlined instead of boldfaced.
> > 3. In Table 2, we will underline the second-best results for consistency with Table 1 and to better highlight the performance margins.
> >
> > ---
> >
> > **Q5:** Figure 3 shows the ablation result on virtual outliers on Far-OOD datasets. What about ablation study on Near-OOD datasets?
> >
> > **A5:** Thank you for pointing it out. We have performed a comprehensive ablation on virtual outliers across different datasets including the Near-OOD datasets.
> > Due to space limitation, only the average results are presented below:
> >
> > |        |          | CIFAR-100         |                 | ImageNet-1k       |                 | ImageNet-1k       |                 |
> > |--------|----------|-------------------|-----------------|-------------------|-----------------|-------------------|-----------------|
> > |        | Backbone | ResNet            |                 | ResNet            |                 | ViT               |                 |
> > |        |          | FPR95↓ | AUROC↑ | FPR95↓ | AUROC↑ | FPR95↓ | AUROC↑ |
> > | w/o virtual outliers | Far OOD  | 28.31             | 92.79           | 42.92             | 85.75           | 33.85             | 91.02           |
> > | w virtual outliers  | Far OOD  | **14.71**         | **96.35**       | **18.91**         | **96.33**       | **16.09**         | **96.67**       |
> > | w/o virtual outliers | Near OOD | 51.32             | 82.57           | 80.3              | 65.38           | 68.15             | 78.08           |
> > | w virtual outliers   | Near OOD | **42.67**         | **86.45**       | **67.83**         | **81.12**       | **59.93**         | **83.72**     |
> >
> > From the results, incorporating virtual outliers consistently improves OOD detection performance, including on Near-OOD datasets. We will include the complete experimental results in the Appendix for a more comprehensive discussion.

---

> > > ### Author Response · Authors · 2024-11-24
> > > **Thank you; invitation to reply**
> > >
> > > Dear reviewer,
> > >
> > > We truly appreciate your time and effort in reviewing our work. We hope our response has addressed your concerns. Please do not hesitate to let us know if you have any further questions or feedback.

---

> > > > ### Comment · Reviewer_jBPb · 2024-11-28
> > > >
> > > > Thank you for the clarification and additional experimental results. These address my questions and I'll keep my score.

---

### Official Review · Reviewer_q2aw · 2024-11-02

**Soundness:** 4
**Presentation:** 3
**Contribution:** 4
**Rating:** 8
**Confidence:** 4

**Summary:**

This paper focuses on detecting Out-of-Distribution (OOD) inputs, which is a crucial task for improving the reliability of deep neural networks. Since there exists an inherent distribution shift between ID and OOD data, the authors propose a novel method Prototype-based Optimal Transport (POT) which utilizes the prototype-based optimal transport to assess the discrepancy between test inputs and ID prototypes and use the obtaining transport costs for OOD detection. Since relying solely on the cost of ID prototypes is insufficient for discerning OOD data with smaller distribution shifts from ID data, they propose to generate virtual outliers to approximate the OOD region using representation linear representation extrapolation. Finally, experiments on several benchmark datasets are provided.

**Strengths:**

1. This paper addresses the OOD detection problem by measuring distribution discrepancy and proposes a novel approach using prototype-based optimal transport.
2. To identify OOD data with smaller distribution shifts from ID data, this paper proposes to generate virtual outliers to approximate the OOD region using representation linear representation extrapolation.
3. Comprehensive experimental results on various benchmark datasets demonstrate the effectiveness of the proposed detection method.

**Weaknesses:**

1. The experimental results and ablation studies require more explanation and discussion, rather than just describing the phenomena.
2. Although this paper reduces computational complexity by introducing the entropy regularization term and the Sinkhorn-Knopp algorithm, the optimal transport problem is still computationally intensive on large-scale datasets. This may limit the application of the proposed method POT in resource-constrained environments.
3. Is the constraint w<0 in Eq. (8) necessary? Because there is no corresponding description in this paper, and this case was not considered in the parameter analysis.

**Questions:**

1.	Can the authors provide more discussions on the experimental results and ablation studies?
2.	Can the authors show the execute time of the proposed method compared with other methods?

---

> ### Author Response · Authors · 2024-11-21
> **Thank you; more discussion  and new execution time experiments [1/2]**
>
> **Q1:** The experimental results and ablation studies require more explanation and discussion, rather than just describing the phenomena. Can the authors provide more discussions on the experimental results and ablation studies?
>
> **A1:** Thanks for your valuable suggestion. We will add more discussions in Section 4.2 Empirical Results and Analysis. Here are the additional discussions to be included:
>
> 1. **Ablation on virtual outliers.** Figure 3 displays the score distribution of test inputs, where the overlap represents the misclassified samples, typically those OOD samples near ID data. Notably, incorporating virtual outliers reduces the overlap, indicating improved separability. However, we also observe that the peaks of the OOD and ID scores may become closer, as shown in Figures 3 (c) and (d).
> We reckon it is due to that the virtual outliers are specifically design to highlight the detection of near OOD samples. But they may slightly compromise the detection of far OOD samples that are inherently easier to distinguish, resulting in closer peaks between OOD and ID scores.
> We conducted experiments with different transport costs as score, and the results are shown below:
>
> | Prototypes    | Virtual Outliers | FPR95↓   | AUROC↑   | FPR95↓   | AUROC↑   |
> |---------------|------------------|----------|----------|----------|----------|
> | ✔             |                  | 28.07    | 92.80    | 33.85    | 91.02    |
> |               | ✔                | 30.67    | 91.43    | 94.12    | 24.92    |
> | ✔             | ✔                | **14.35**| **96.70**| **15.59**| **96.78**|
>
> From the results, using ID prototypes or virtual outliers alone fails to achieve consistent detection performance. This suggests that the transport cost to ID prototypes and virtual outliers may focus on detecting different parts of OOD data, complementing each other to collectively enhance overall detection performance.
>
> 2. **Parameter analysis.**
>
>     (1) As shown in Figure 4(a), increasing the test batch size improves OOD detection performance. We conjecture that larger batches result in the empirical distribution (i.e., the observed data distribution) better approximating the real test data distribution, enabling a more accurate measurement of distribution differences.
>
>     (2) In Figure 4(b), the OOD detection performance converges as the entropic regularization coefficient λ increases. This occurs because the entropy term gradually dominates the optimization of OT with λ increasing, eventually leading to a uniform transport plan with the maximum entropy. As a result, the OOD detection performance stabilizes.
>
>     (3) In Figure 4(c), OOD detection performance improves initially and then declines as the extrapolation coefficient ω increases. We speculate this is because the generated virtual outliers progressively move from the ID prototypes towards the OOD region as ω increases. When ω is too small, the virtual outliers remain close to the ID data, whereas large ω places the virtual outliers far from the OOD data, both leading to suboptimal performance.

---

> > ### Author Response · Authors · 2024-11-21
> > **Thank you; more discussion and new execution time experiments [2/2]**
> >
> > **Q2:** Although this paper reduces computational complexity by introducing the entropy regularization term and the Sinkhorn-Knopp algorithm, the optimal transport problem is still computationally intensive on large-scale datasets. This may limit the application of the proposed method POT in resource-constrained environments.
> > Can the authors show the execute time of the proposed method compared with other methods?
> >
> > **A2:** Thank you for raising the concern. Our method is computationally efficient and takes short time for OT calculations, ensuring practicality in real-world applications.
> > We selected the top-3 performing methods as baselines and compared them with POT in terms of execution time on the large-scale ImageNet benchmark:
> >
> > | Dataset | SSB-hard |            | NINCO |            | iNaturalist |            | Textures |            | OpenImage-O |            |
> > |--------|----------|------------|-------|------------|-------------|------------|---------|------------|-------------|------------|
> > |        | time(s)↓ | AUROC↑     | time(s)↓ | AUROC↑     | time(s)↓    | AUROC↑     | time(s)↓ | AUROC↑     | time(s)↓    | AUROC↑     |
> > | GEN    | **785.76**  | 70.19      | **640.65** | 82.47      | **797.52**  | 93.53      | 608.08  | 90.25      | 695.13      | 90.25      |
> > | DDE    | -        | 76.29      | -     | 81.32      | -           | 97.10      | -       | 88.96      | -           | 88.97      |
> > | NAC    | 859.73   | 68.04      | 735.71 | 82.45      | 821.94      | 93.69      | 728.96  | 94.17      | 775.51      | 91.54      |
> > | POT    | 998.02   | **80.35**  | 702.38 | **87.62**  | 930.79      | **99.41**  | **423.96** | **95.36** | **493.51**  | **95.57**  |
> >
> > Due to all the execution times of DDE exceeding 2 hours, we excluded it from the comparison. This prolonged execution time arises because DDE requires training a DNN for each test batch, with the majority of the time consumed by the iterative training process.
> > From the exhibited results, POT achieves superior performance while maintaining high efficiency.
> >
> > The execution time of POT primarily comprises two components: feature extraction and optimal transport (OT) computation.
> > For a more detailed view, we provided the time for OT calculation below, which takes only about 10 seconds per dataset:
> >
> > |Dataset          | SSB-hard | NINCO | iNaturalist | Textures | OpenImage-O |
> > |------------------|----------|-------|-------------|----------|-------------|
> > | number of samples| 49000    | 5879  | 10000       | 5160     | 15869       |
> > | Calculation Time (s) | 12.31  | 7.27  | 8.51        | 7.92     | 8.99        |
> >
> > These results further demonstrate the efficiency of our method, underscoring its feasibility and potential for real-world deployment.
> >
> > ---
> >
> > **Q3:** Is the constraint ω < 0 in Eq. (8) necessary? Because there is no corresponding description in this paper, and this case was not considered in the parameter analysis.
> >
> > **A3:** No, the constraint $\omega > 1 \vee \omega < 0$ in Eq. (8) is primarily to keep the equation intact, since linear extrapolation can theoretically be applied in two opposite directions: beyond the ID prototypes or applied to the mean of the test data. However, in our work, we focus exclusively on generating virtual outliers beyond the mean of the test data. Therefore, only the corresponding constraint $\omega > 1$ is considered and discussed in the remainder of the paper.

---

> ### Comment · Reviewer_q2aw · 2024-11-23
> **To response**
>
> All my concerns have been addressed. I will raise my score.

---

> > ### Author Response · Authors · 2024-11-24
> > **Thank you so much**
> >
> > Dear reviewer,
> >
> > We are glad to hear that our response has addressed your concerns. Your comments have significantly contributed to improving our paper. We sincerely appreciate your time and effort in reviewing our work!

---

### Official Review · Reviewer_49JF · 2024-11-03

**Soundness:** 2
**Presentation:** 2
**Contribution:** 3
**Rating:** 3
**Confidence:** 5

**Summary:**

Summary:This paper proposes a new prototype-based optimal transfer (POT) method to detect out-of-distribution (OOD) data by measuring the optimal total transfer cost between the test sample and the ID prototype and the virtual outlier. In addition, this paper also proposes to obtain the virtual outlier of OOD by simple linear extrapolation, and demonstrates its effectiveness by comparing with the existing technology. The paper has a clear structure and rigorous logic, covering relevant background, methods, experimental evaluation and results analysis.

**Strengths:**

- This paper provides a new perspective on the OOD detection problem by combining the concepts of prototype and optimal transmission.
- The author conducted exhaustive experiments on multiple standard datasets, comparing the performance of POT with 21 existing technologies. The experiments are adequately designed and the results are clearly interpreted.

**Weaknesses:**

- Formulas 2 and 3 in the paper mention that the calculation of the transmission plan and cost matrix comes from the ID prototype and test samples. The simple linear extrapolation to obtain the OOD area is also based on the test samples. This solution is not realistic under the task setting of OOD detection.
- The paper mentions that the OOD area is obtained by simple linear extrapolation, but based on the author's description and existing research, the difficulty of OOD samples lies in the approximate ID features in the feature space. The OOD area obtained by simple linear extrapolation may not be representative, and there is a lack of discussion on the interpretability of the process.
- The results of the paper method should also be compared with the results of some of the latest papers.

**Questions:**

- Are you skeptical about the effectiveness of obtaining OOD data areas through simple linear extrapolation?
- Is it realistic for OOD detection to use test data for linear extrapolation?

---

> ### Author Response · Authors · 2024-11-21
> **Thank you; new experiments and justifications**
>
> **Q1:** The simple linear extrapolation to obtain the OOD area is also based on the test samples. Is it realistic for OOD detection to use test data for linear extrapolation?
>
> **A1:**
> Thank you for raising this concern. We would like to argue that our solution is both practical and feasible in real-world applications, for the following reasons:
>
> 1. Our method does not require any label information about test samples.
> 2. Our method only utilizes the empirical mean of test inputs to help generate virtual outliers, and such information can be easily collected in the test phase.
> 3. Utilizing the empirical mean of test inputs is a common practice in data processing. For instance, empirical mean and variance of batch data are used in batch normalization layers of DNNs to mitigate internal covariate shift [R1].
>
> [R1] Batch Normalization: Accelerating Deep Network Training by Reducing Internal Covariate Shift. ICML. 2015
>
> ---
>
> **Q2:** The OOD area obtained by simple linear extrapolation may not be representative, and there is a lack of discussion on the interpretability of the process. Are you skeptical about the effectiveness of obtaining OOD data areas through simple linear extrapolation?
>
> **A2:**
> We would like to justify the effectiveness of linear extrapolation from the following aspects:
>
> 1. **Interpretable generation process of virtual outliers.** As detailed in our paper, the mean of test data $M$ is derived as a linear interpolation of the means of test ID samples $ {M} _ {in}$ and test OOD samples $M _ {out}$. With the ID prototypes $P$ constructed as the ID sample mean ${M} _ {in}$, linear extrapolation between $M$ and $P$ effectively captures the information about the underlying distribution of test OOD data. This process guides the generation of virtual outliers, which are strategically positioned in the transitional region between ID and OOD data. By tuning the extrapolation coefficient, our method adjusts the placement of virtual outliers to approximate OOD data near the ID distribution, thereby emphasizing their detection.
> 2. **Comparison with existing approaches.** While other OOD detection methods [R1, R2] also leverage extrapolation or interpolation to generate outliers, they typically rely on auxiliary OOD datasets to produce a large volume of novel outliers, attempting to extend the detectable OOD distribution. However, it is theoretically infeasible to cover the vast potential space of OOD samples. In contrast, our approach eliminates the need for auxiliary datasets and instead focuses on generating a small number of representative virtual outliers tailored to the test OOD data, thereby achieving both efficiency and effectiveness.
> 3. **Experimental validation.** We performed a comprehensive ablation on linear extrapolation (LE) across all the benchmarks. Due to space limitations, only the average results are presented below:
>
> |   |  | CIFAR-100         |     | ImageNet-1k       |       | ImageNet-1k    |    |
> |--------|----------|-------------------|-----------------|-------------------|-----------------|-------------------|-----------------|
> |        | Backbone | ResNet            |                 | ResNet            |                 | ViT               |                 |
> |        |          | FPR95$\\downarrow$ | AUROC$\\uparrow$ | FPR95$\\downarrow$ | AUROC$\\uparrow$ | FPR95$\\downarrow$ | AUROC$\\uparrow$ |
> | w/o LE | Far OOD  | 28.31             | 92.79           | 42.92             | 85.75           | 33.85             | 91.02           |
> | w LE   | Far OOD  | **14.71**         | **96.35**       | **18.91**         | **96.33**       | **16.09**         | **96.67**       |
> | w/o LE | Near OOD | 51.32             | 82.57           | 80.3              | 65.38           | 68.15             | 78.08           |
> | w LE   | Near OOD | **42.67**         | **86.45**       | **67.83**         | **81.12**       | **59.93**         | **83.72**     |
>
> The results consistently show that incorporating linear extrapolation significantly improves OOD detection performance.
>
> [R1] Diversified outlier exposure for out-of-distribution detection via informative extrapolation. ICLR. 2023 \
> [R2] Mixture Outlier Exposure: Towards Out-of-Distribution Detection in Fine-grained Environments. WACV. 2023
>
> ---
>
> **Q3:** The results of the paper method should also be compared with the results of some of the latest papers.
>
> **A3:** Thank you for this valuable suggestion. We have already compared our method with the latest post-hoc OOD detection method, NAC-UE [R3], as well as the most recent baseline that does not require training data, GEN [R4]. If there are additional methods you believe we should include in our comparison, please let us know promptly so we can present the experiment results.
>
> [R3] Neuron activation coverage: Rethinking out-of-distribution detection and generalization. ICLR, 2024. \
> [R4] GEN: pushing the limits of softmax-based out-of-distribution detection. CVPR, 2023.

---

> > ### Author Response · Authors · 2024-11-24
> > **Thank you; invitation to reply**
> >
> > Dear reviewer,
> >
> > We truly appreciate your time and effort in reviewing our work. We hope our response has addressed your concerns. Please do not hesitate to let us know if you have any further questions or feedback.

---

### Comment · Area_Chair_jCvq · 2024-11-22
**Interactive Discussions**

Dear Reviewers,

Thank you for your efforts in reviewing this paper. We highly encourage you to participate in interactive discussions with the authors before November 26, fostering a more dynamic exchange of ideas rather than a one-sided rebuttal.

Please feel free to share your thoughts and engage with the authors at your earliest convenience.

Thank you for your collaboration.

Best regards,
ICLR 2025 Area Chair

---

### Meta-Review · Area_Chair_jCvq · 2024-12-19

**Metareview:**

This submission focuses on out-of-distribution (OOD) detection and proposes a novel method based on prototypes and optimal transport. Specifically, the method first calculates the optimal transport distance from each test batch to the in-distribution (ID) prototypes. To improve performance, the distance to virtual outliers, generated through linear extrapolation, is then incorporated into the OOD detection score.

One key weakness highlighted by reviewer #7P4j concerns the problem setting, where the proposed method relies on a minibatch of test samples to obtain the OOD detection scores. This approach is impractical for real-world applications and differs from existing OOD detection methods. Additionally, other reviewers, such as #zWXM, expressed similar concerns. The Area Chair concurs with these points and suggests that the authors address this issue in a future submission.

**Additional Comments On Reviewer Discussion:**

As mentioned above, the main concern is the unfair comparison with existing OOD detection methods, as the proposed approach requires a minibatch of test samples to calculate the OOD scores for each sample. The effects of batch size or the distribution of OOD samples within the minibatch remain unclear. The rebuttal does not adequately address this issue.

Another unresolved issue is the lack of explanation as to why optimal transport (OT) methods perform well for OOD detection, as well as the justification for using linearly extrapolated virtual OOD outliers.

---

### Decision · Program_Chairs · 2025-01-22

Reject